# SPRINT: Stochastic Performative Prediction With Variance Reduction

## Abstract

Performative prediction (PP) is an algorithmic framework for optimizing machine learning (ML) models where the model's deployment affects the distribution of the data it is trained on. Compared to traditional ML with fixed data, designing algorithms in PP converging to a stable point – known as a stationary performative stable (SPS) solution – is more challenging than the counterpart in conventional ML tasks due to the model-induced distribution shifts. While considerable efforts have been made to find SPS solutions using methods such as repeated gradient descent (RGD) and greedy stochastic gradient descent (SGD-GD), most prior studies assumed a strongly convex loss until a recent work established $\mathcal{O}(1/\sqrt{T})$ convergence of SGD-GD to SPS solutions under smooth, non-convex losses. However, this latest progress is still based on the restricted bounded variance assumption in stochastic gradient estimates and yields convergence bounds with a non-vanishing error neighborhood that scales with the variance. This limitation motivates us to improve convergence rates and reduce error in stochastic optimization for PP, particularly in non-convex settings. Thus, we propose a new algorithm called stochastic performative prediction with variance reduction (SPRINT) and establish its convergence to an SPS solution at a rate of $\mathcal{O}(1/T)$. Notably, our rate removes the explicit bounded-variance assumption and contains no variance parameters. Experiments on multiple real datasets with non-convex models demonstrate that SPRINT outperforms SGD-GD in both convergence rate and stability.

## 1 Introduction

Machine learning (ML) models are typically trained under the assumption that both training and testing data are drawn independently from a static distribution. However, in many real-world applications where an ML system is used to make predictions about humans, this assumption is often violated, as the deployment of the model can alter the behavior of the population interacting with it, thereby inducing changes in the data distribution that the model is meant to predict. This phenomenon, known as "performative effects" or "model-induced distribution shift" (Perdomo et al., 2021; Drusvyatskiy & Xiao, 2023), presents distinctive challenges to conventional learning paradigms. For example, in strategic classification scenarios, individuals can manipulate their features (e.g., by modifying their resumes or financial profiles) to obtain more favorable results (Hardt et al., 2015); In digital platforms, users may change their engagement behavior based on their beliefs about how the model operates (Zhang et al., 2019; Chi et al., 2022); In adversarial settings such as spam detection, spammers continuously adapt their tactics to evade newly deployed filters. Understanding and mitigating these performative effects is essential for building robust ML models.

Thus, Perdomo et al. (2021) introduced performative prediction (PP) as the first optimization framework for model-dependent distribution shifts. In this framework, the data distribution $\mathcal{D}(\boldsymbol{\theta})$ is explicitly modeled as a mapping from the ML model parameters $\boldsymbol{\theta}$ to the space of distributions, and the goal is to minimize risk for this distribution. Since the data distribution itself depends on the model being optimized, the objective becomes minimizing the performative risk (PR), i.e.,

$$\boldsymbol{\theta}^{\text{PO}} = \operatorname*{argmin}_{\boldsymbol{\theta}} \mathcal{V}(\boldsymbol{\theta}) \overset{\text{def}}{=} \mathbb{E}_{z \sim \mathcal{D}(\boldsymbol{\theta})}[\ell(z; \boldsymbol{\theta})], \tag{1}$$

where $\ell(\boldsymbol{\theta}; z)$ is the loss function and $z = (x, y)$ is sampled from the distribution $\mathcal{D}(\boldsymbol{\theta})$. The coupling nature of PP implies that $\boldsymbol{\theta}$ determines both the model and the data distribution, and the

minimizer $\theta^{\mathrm{PO}}$ is referred to as the **performative optimal** (PO) solution. Since the data distribution is itself a function of the variable $\theta$ being optimized, finding $\theta^{\mathrm{PO}}$ is often intractable (Perdomo et al., 2021; Izzo et al., 2021). To address this challenge, instead of finding PO solution, many existing works focused on **performative stable** (PS) solution $\theta^{\mathrm{PS}}$ to minimize the **decoupled performative risk** $\mathcal{J}(\theta; \theta^{\mathrm{PS}})$ in the following form:

$$\theta^{\mathrm{PS}} = \operatorname*{argmin}_{\theta} \mathcal{J}(\theta; \theta^{\mathrm{PS}}) \stackrel{\text{def}}{=} \mathbb{E}_{z \sim \mathcal{D}(\theta^{\mathrm{PS}})}[\ell(z; \theta)]. \tag{2}$$

Different from $\mathcal{V}(\theta)$ where the data distribution $\mathcal{D}(\theta)$ depends on the variable $\theta$ being optimized, $\mathcal{J}(\theta; \theta^{\mathrm{PS}})$ decouples the two: the data distribution is induced by a fixed parameter $\theta^{\mathrm{PS}}$, and the the learner only needs to minimize the loss over this fixed distribution. This enables us to find $\theta^{\mathrm{PS}}$ iteratively using *repeated optimization* schemes—by repeatedly updating $\theta_{t+1}$ to minimize risk on a fixed distribution $\mathcal{D}(\theta_t)$. If the procedure converges, then the minimizer $\theta_{t+1}$ approaches $\theta^{\mathrm{PS}}$. Examples of such approaches commonly used in the literature include repeated risk minimization (Perdomo et al., 2021), repeated gradient descent (Perdomo et al., 2021), and stochastic gradient descent-greedy deploy (SGD-GD) (Mendler-Dünner et al., 2020). Since the population size can be large in practice, it is not feasible to repeatedly run full gradient descent at each iteration (Perdomo et al., 2021). In this paper, we consider stochastic optimization in PP with a primary focus on the SGD-GD approach (Li & Wai, 2024) since it is more computationally efficient and requires now knowledge of the performative phenomena. In general, the SGD-GD approach can be written as in the following form:

$$\theta_{t+1} = \theta_t - \gamma_{t+1} \nabla \ell(z_{t+1}; \theta_t), \ z_{t+1} \sim \mathcal{D}(\theta_t). \tag{3}$$

It has been shown that Eqn. 3 could converge to a unique PS solution (Mendler-Dünner et al., 2020), but under two strong assumptions: (i) the loss function $\ell$ is strongly convex and smooth in $\theta$; and (ii) the sensitivity of the distribution map $\mathcal{D}$ measured by the Wasserstein-1 distance is upper bounded. However, the strong convexity requirement significantly limits the practical applicability of PP. A very recent work (Li & Wai, 2024) addressed this limitation by establishing the first convergence guarantees to a stationary performative stable (SPS) solution (cf. Def. 1.1) under the more general setting where $\ell$ is smooth and non-convex.

**Definition 1.1** ($\delta$-Stationary Performative Stable Solution (Li & Wai, 2024))**.** For a given $\delta \geq 0$, the vector $\theta_{\delta\text{-SPS}} \in \mathbb{R}^d$ is said to be a $\delta$-stationary performative stable ($\delta$-SPS) solution of Eqn. 2 if

$$\left\| \nabla \mathcal{J}(\theta_{\delta\text{-SPS}}; \theta_{\delta\text{-SPS}}) \right\|^2 = \left\| \mathbb{E}_{z \sim \mathcal{D}(\theta_{\delta\text{-SPS}})} \left[ \nabla \ell(z; \theta_{\delta\text{-SPS}}) \right] \right\|^2 \leq \delta.$$

The parameter $\delta$ in Def. 1.1 measures the stability of a solution. When $\delta = 0$, $\theta_{\delta\text{-SPS}}$ coincides with PS solution $\theta^{\mathrm{PS}}$. As showed by Li & Wai (2024), SGD-GD can converge to $\theta_{\delta\text{-SPS}}$ at a rate of $\mathcal{O}(1/\sqrt{T})$ for non-convex, smooth loss function $\ell$ under an additional assumption of **bounded variance in the stochastic gradient estimates**. Moreover, the convergence bound includes a *non-vanishing* error neighborhood that *scales with the variance*. This naturally motivates the following fundamental question:

> **(Q)**: Is it possible to control the stochastic gradient estimate variance in PP to develop a new SGD-GD-type algorithm, which converges to $\theta_{\delta\text{-SPS}}$ i) with a faster convergence rate and ii) an error neighborhood *independent* of the gradient variance? If yes, how?

We note that, in the literature, several variance reduction techniques have been developed for the conventional (i.e., non-PP) stochastic gradient descent (SGD) method in ML (e.g., the stochastic average gradient method (SAG) (Roux et al., 2012), the stochastic average gradient augmented (SAGA) method (Defazio et al., 2014), the stochastic variance reduced gradient method (SVRG) (Reddi et al., 2016), and the stochastic path-integrated differential estimator (SPIDER) (Fang et al., 2018)), which achieve better convergence rates. However, their applicability and effectiveness in the PP settings remain largely *unknown*. In light of the increasing importance of PP and its significant gap between the theory and practice, our goal in this paper is to develop new variance-reduced SGD-GD-based algorithms for PP.

As a starting point in the new variance-reduced PP paradigm, we propose a stochastic performative prediction with variance reduction (SPRINT) method, which is the first variance-reduced performative prediction (PP) framework and is inspired by the SVRG approach(Reddi et al., 2016). Our key

idea is to divide the updating iterations of SGD-GD into multiple epochs and store a snapshot of the full gradient $\nabla \mathcal{J}$ at the end of each epoch. At each iteration of repeated optimization, the most recent snapshot can reduce the variance of the current stochastic gradient descent with a slight bias.

We note, however, that our proposed SPRINT approach is *far from* a straightforward application of SVRG in the PP paradigm. Unlike conventional stochastic optimization, the unique nature of PP settings introduces two major challenges in algorithm design and theoretical analysis: (i) The effects change the data distribution and the resulting decoupled performative loss gradient across iterations, thereby rendering the full gradient snapshot taken in the previous iteration a *biased* estimate of the current stochastic gradient evaluated at the same $\boldsymbol{\theta}$ on a different data distribution; and (ii) The effects introduce new difficulties, absent in the standard SVRG theoretical analysis, in constructing a suitable Lyapunov function and establishing a negative Lyapunov drift, since the change from $\boldsymbol{\theta}_k^s$ to $\boldsymbol{\theta}_{k+1}^s$ incurs extra distribution shifts unseen in the non-PP settings.

The major contribution of this paper is that we overcome the above technical challenges and establish the convergence of our SPRINT method to an SPS solution with a *variance-independent* fast convergence rate. The main results and key contributions of this work are summarized as follows:

- We propose the SPRINT algorithm to reduce the variance of the SGD-GD-Based approach in non-convex PP settings. Compared to existing works, SPRINT converges to an SPS solution at an *accelerated* rate of $\mathcal{O}(1/T)$, and, more importantly, features an error neighborhood that is **independent** of the stochastic gradient variance. To our knowledge, all these results are first in the literature.

- To establish the aforementioned $\mathcal{O}(1/T)$ convergence rate of our SPRINT method, we propose a series of new Lyapunov function construction techniques, which are of general and independent interests in the PP literature. Moreover, we derive the incremental first-order oracle (IFO) complexity (the number of operations of taking a sample and calculating its gradient to achieve a $\mathcal{O}(\delta)$-SPS (Reddi et al., 2016)) of the algorithm.

- We conduct extensive numerical experiments to validate the theoretical results on three real-world datasets. In addition to the credit data (Kaggle, 2012) and MNIST (Deng, 2012), we present additional experimental results for training MLP/CNN models using SGD-GD-Based and SPRINT approaches on the full CIFAR-10 (Krizhevsky et al., 2009) dataset. These broad experiments verify and enrich the practical applicability of optimization-based approaches for PP.

The remainder of this paper is organized as follows: In Sec. 2, we review the related literature and formally present the problem formulation along with a discussion of prior results in Sec. 3. We then present our SPRINT algorithm in Sec. 4 and prove its convergence and IFO complexity in Sec. 5. Lastly, we present the numerical results in Sec. 6 and conclude the paper in Sec. 7.

## 2 RELATED WORK

In this section, we provide a brief overview on three research areas that are closely related to our work: 1) finding SPS solutions for PP; 2) other related solution metrics in PP; and 3) variance reduction (VR) techniques for SGD-based optimization methods.

**1) Finding PS/SPS solutions for PP:** Performative prediction (PP) was first formulated as an optimization framework by Perdomo et al. (2021) to handle endogenous data distribution shifts, based on which an iterative optimization procedure named *repeated risk minimization* (RRM) to find a performative stable point and also bound the distance between the PS solution and the PO solution in (Perdomo et al., 2021). Perdomo et al. (2021) also proposed a repeated gradient descent (RGD) method, but a full gradient is required in each iteration. In contrast, Mendler-Dünner et al. (2020) designed the first algorithm to find the PS solution under the online setting. Mendler-Dünner et al. (2022) later established the convergence rate of greedy deployment and lazy deployment after each random update under the assumptions of smoothness and strong convexity. Later, Mofakhami et al. (2023) investigated training neural network in the PP settings, but they assumed the model as $\hat{y} = f_{\boldsymbol{\theta}}$. The loss function is written as $\ell(\hat{y}, y)$ and assumed to be strongly convex to $\hat{y}$. The distribution map $D(\hat{y})$ must also be $\epsilon$-sensitive to $\hat{y}$. Most recently, Li & Wai (2024) provided the first convergence results of the PP settings where the loss function in PP is non-convex and no particular solution structure is assumed. However, this result is based on the bounded variance assumption

limitation and suffers from a non-vanishing error neighborhood that scales linearly with the variance. Our work overcomes this limitation by proposing a variance reduction approach to accelerate the convergence and mitigate the error neighborhood.

**2) Related solution metrics for PP:** It is worth noting that, besides finding SPS solutions, there also exist other solution metrics for PP. Miller et al. (2021) tackled PP problems by directly optimizing performance risk (PR) to find the PO solution for a restricted set of the distribution maps. Ray et al. (2022); Liu et al. (2023) utilized derivative-free optimization to find PO solutions. Izzo et al. (2021) also designed an algorithm based on performative gradient descent for finding PO solutions under a convex PR assumption, which, however, is hard to verify even with a strongly convex loss function. Zhu et al. (2023); Zheng et al. (2024) also studied PP under weakly convex conditions. Brown et al. (2022); Li & Wai (2022) focused on state-dependent PP settings and proposed algorithms that converge to PS solutions. There also exist other works on PP that focused on fairness(Jin et al., 2024), social welfare (Kim & Perdomo, 2022) and privacy (Li et al., 2024) metrics.

**3) Variance reduction techniques for SGD-based optimization:** In the literature, variance reduction (VR) techniques have been developed to accelerate the convergence of the SGD method (Roux et al., 2012; Reddi et al., 2016; Nguyen et al., 2017; Defazio et al., 2014; Fang et al., 2018). For example, stochastic average gradient (SAG) (Roux et al., 2012) and stochastic average gradient augmented (SAGA) (Defazio et al., 2014) maintain an average of the stochastic gradients of all data samples with infrequent updates. However, these approaches could incur high memory costs in the large dataset regime. In contrast, the stochastic variance-reduced gradient (SVRG) (Reddi et al., 2016) eliminates the need for storing per-sample stochastic gradients by periodically computing a full gradient at the beginning of each epoch and using it to construct a variance-reduced update. SVRG enjoys a linear convergence rate with a modest memory requirement under the strong convexity setting. Subsequently, the stochastic recursive gradient (SARAH) (Nguyen et al., 2017) and the stochastic path-integrated differential estimator (SPIDER)(Fang et al., 2018) methods further improve SVRG by incorporating fresher recursive iterates to achieve optimal convergence rate in terms of both stationary gap and dataset size dependencies. We note, however, that all these VR techniques were only designed for non-PP settings. To date, the development of VR techniques for PP remains largely underexplored in the literature.

**4) Reinforcement Learning and Optimal Control:** While Reinforcement Learning (RL) (Kaelbling et al., 1996) optimizes the reward brought by policy $\pi$ $J(\pi) = \mathbb{E}_{\tau \sim p_\pi}[R(\tau)]$ directly under a distribution $p_\pi$, it is reward-based and does not seek an explicit equilibrium between the policy and the induced distribution. PP defines a stable solution satisfying $\|\nabla_\theta J(\theta; \theta)\| \leq \delta$, which enforces *equilibrium between the model parameters and the data distribution they induce. For the different optimization algorithms their convergence in RL settings, we refer to Xiao (2022); Beggs (2005); Agarwal & Agarwal; Yuan et al. (2022).

For Optimal Control (OC) (Todorov et al., 2006), it studies dynamic state transitions $x_{t+1} = f(x_t, u_t)$ and minimizes cumulative cost $\sum_t c(x_t, u_t)$. The dependency is primarily temporal, while in PP the dependency is model-dependent and we still aim to find an equilibrium. Optimal control relies on Bellman recursion and dynamic programming, while our work instead analyzes convergence to a stationary equilibrium via Wasserstein continuity and Lyapunov drift arguments.

## 3 PRELIMINARIES AND STATE OF THE ART OF PERFORMATIVE PREDICTION (PP)

Following the convention in the variance reduction literature (e.g., (Reddi et al., 2016; Roux et al., 2012)), we assume the full population contains $n$ samples $\{z_1, ..., z_n\} \subset \mathcal{Z}$ in total [1] and the ML model is parameterized by $\boldsymbol{\theta} \in \Theta \subseteq \mathbb{R}^d$. Then, the expected gradient of the decoupled performative risk among the full population can be computed as: $\nabla \mathcal{J}(\boldsymbol{\theta}; \boldsymbol{\theta}) = \mathbb{E}_{z \sim \mathcal{D}(\boldsymbol{\theta})}[\nabla \ell(\boldsymbol{\theta}; z)] = \frac{1}{n} \sum_{i=1}^{n} \nabla \ell(\boldsymbol{\theta}; z_i)$, where $\nabla \ell(\boldsymbol{\theta}; z)$ denotes the gradient of $\ell(\boldsymbol{\theta}; z)$ with respect to $\boldsymbol{\theta}$. Unlike most prior works that studied the convergence of SGD-GD (Eqn. 3) in performative prediction (PP) under

---

[1]This finite sum setting has been widely used in variance reduction literature (Roux et al., 2012; Reddi et al., 2016; Nguyen et al., 2017) and is practical in PP settings because typical applications (e.g., credit scoring (Perdomo et al., 2021), college admission (Xie et al., 2024)) often involve a finite population. Moreover, our framework can be extended to infinite sum settings as discussed in App. G.

the assumption that the loss function $\ell$ is strongly convex (Mendler-Dünner et al., 2020; Izzo et al., 2021; Mofakhami et al., 2023), we consider *non-convex* $\ell$ and focus on the problem setup studied by Li & Wai (2024), a recent work that showed the convergence of SGD-GD to an SPS solution in non-convex PP settings. To familiarize the readers with necessary background and facilitate the subsequent discussions, we first present the technical assumptions and then the theoretical results of (Li & Wai, 2024). Throughout the paper, we use $\|\cdot\|$ to denote $\ell_2$ norm.

**Assumption 3.1** (Smoothness and Lower Bound of the Gradients). For any $z \in \mathcal{Z}$, there exists a constant $L, L_0 \geq 0$ such that

$$\|\nabla\ell(z;\boldsymbol{\theta}) - \nabla\ell(z;\boldsymbol{\theta'})\| \leq L\|\boldsymbol{\theta} - \boldsymbol{\theta'}\|, \quad \forall\boldsymbol{\theta},\boldsymbol{\theta'} \in \mathbb{R}^d$$
$$\|\nabla\ell(z;\boldsymbol{\theta}) - \nabla\ell(z';\boldsymbol{\theta})\| \leq L_0\|z - z'\|, \quad \forall z,z' \in \mathcal{Z}$$

Moreover, there exists a constant $\ell^\star > -\infty$ such that $\ell(z;\boldsymbol{\theta}) \geq \ell^\star$ for all $\boldsymbol{\theta}, z$.

**Definition 3.2** ($\epsilon$-Sensitivity). Distribution map $\mathcal{D}(\theta)$ is $\epsilon$-sensitive if there exists an $\epsilon \geq 0$ such that, for any $\boldsymbol{\theta}, \boldsymbol{\theta'} \in \mathbb{R}^d$, we have

$$W_1(\mathcal{D}(\boldsymbol{\theta}), \mathcal{D}(\boldsymbol{\theta'})) \leq \epsilon\|\boldsymbol{\theta} - \boldsymbol{\theta'}\|,$$

where $W_1(\cdot, \cdot)$ is Wasserstein-1 distance (Li & Wai, 2024).

**Assumption 3.3** (Lipschitzness). The loss function $\ell$ is $L_0$-lipschitz:

$$|\ell(z;\boldsymbol{\theta}) - \ell(z';\boldsymbol{\theta})| \leq L_0\|z - z'\|, \quad \forall z,z' \in \mathcal{Z}, \boldsymbol{\theta} \in \mathbb{R}^d.$$

**Assumption 3.4** (Variance of Stochastic Gradient Estimates). The stochastic gradient is unbiased. i.e., $\nabla\mathcal{J}(\boldsymbol{\theta_1};\boldsymbol{\theta_2}) = \mathbb{E}_{Z\sim\mathcal{D}(\boldsymbol{\theta_2})}[\nabla\ell(Z;\boldsymbol{\theta_1})]$. Also, there exist constants $\sigma_0, \sigma_1 \geq 0$, such that

$$\mathbb{E}_{Z\sim\mathcal{D}(\boldsymbol{\theta_2})}\left[\|\nabla\ell(Z;\boldsymbol{\theta}_1) - \nabla\mathcal{J}(\boldsymbol{\theta}_1;\boldsymbol{\theta}_2)\|^2\right] \leq \sigma_0^2 + \sigma_1^2\|\nabla\mathcal{J}(\boldsymbol{\theta}_1;\boldsymbol{\theta}_2)\|^2.$$

Note that smoothness (Assumption 3.1) and $\epsilon$-sensitivity defined by Def. 3.2 are standard in the machine learning optimization literature on PP (e.g., (Perdomo et al., 2021; Mendler-Dünner et al., 2020)), while the Lipschitzness of loss function (Assumption 3.3) and the bounded variance assumption (Assumption 3.4) are specific to non-convex PP settings but are still common in optimization literature. Under above assumptions, Li & Wai (2024) proved the following convergence result:

**Lemma 3.5** (Convergence of SGD-GD (Li & Wai, 2024)). *Consider iterative updates of SGD-GD given in Eqn. 3, the following holds for any $T > 1$:*

$$\sum_{t=0}^{T-1}\frac{\gamma_{t+1}}{4}\mathbb{E}\left[\|\nabla J(\boldsymbol{\theta}_t;\boldsymbol{\theta}_t)\|^2\right] \leq \Delta_0 + L_0\epsilon\left(\sigma_0 + (1+\sigma_1^2)L_0\epsilon\right)\sum_{t=0}^{T-1}\gamma_{t+1} + \frac{L}{2}\sigma_0^2\sum_{t=0}^{T-1}\gamma_{t+1}^2, \quad (4)$$

*where $\gamma_t$ is the learning rate at iteration $t$, and $\Delta_0 = \mathcal{J}(\boldsymbol{\theta_0};\boldsymbol{\theta_0}) - \ell_*$ is an upper bound of the initial optimality gap between the initial performative risk and the optimal performative risk $\ell_*$. If one sets $\gamma_t = 1/\sqrt{T}$ for all $t$, the bound in 4 can be reduced to:*

$$\mathbb{E}\left[\|\nabla J(\boldsymbol{\theta}_\tau;\boldsymbol{\theta}_\tau)\|^2\right] \leq 4 \underbrace{\left(\Delta_0 + \frac{L}{2}\sigma_0^2\right)}_{\text{Variance}-\text{Dependent}} \cdot\frac{1}{\sqrt{T}} + 4L_0\epsilon\underbrace{(\sigma_0 + (1+\sigma_1^2)L_0\epsilon)\}}_{\text{Variance}-\text{Dependent}}. \quad (5)$$

Lemma 3.5 shows that SGD-GD converges at a rate of $\mathcal{O}(1/\sqrt{T})$ with an error neighborhood of $\mathcal{O}(4L_0\epsilon(\sigma_0+(1+\sigma_1^2)L_0\epsilon))$. However, this term is directly *influenced by the variance* of the stochastic gradient estimate. Additionally, the convergence rate is affected by the third term on the right-hand side of Eqn. 4, which is also *variance-dependent*. These limitations of the state-of-the-art in PP motivate us to propose a new algorithm for **non-convex** PP that achieves faster convergence to SPS solutions and develop new variance reduction techniques to enable the SGD-GD-Based PP approach to be **variance-independent**.

## 4    THE PROPOSED SPRINT ALGORITHM

| Literature | Loss function $\ell$ | Convergence rate | Error neighborhood | Comments |
|---|---|---|---|---|
| Mendler-Dünner et al. (2020) | Strongly convex | $\mathcal{O}(\frac{1}{T})$ | None | No error due to strong convexity |
| Li & Wai (2024) | Nonconvex | $\mathcal{O}(\frac{1}{\sqrt{T}})$ | $\mathcal{O}(\sigma_0\epsilon + \sigma_1^2\epsilon^2)$ | Variance-dependent error neighborhood |
| This work: SPRINT | Nonconvex | $\mathcal{O}(\frac{1}{T})$ | $\mathcal{O}(\epsilon^2 + \epsilon^4)$ | **Variance-independent** error neighborhood |

Table 1: SPRINT compared to previous literature using SGD-GD. Since the main objective of the paper is to improve the existing SGD-GD-Based PP approaches, we do not compare our results with algorithms that focused on PO solutions (Izzo et al., 2021; Miller et al., 2021) or those using repeated risk minimization (RRM) (Perdomo et al., 2021; Mofakhami et al., 2023). For these methods, we refer readers to Table 1 of Li & Wai (2024) for detailed comparisons.

To address the aforementioned limitations, we propose a new algorithm called SPRINT for PP, which is inspired by the stochastic variance reduced gradient (SVRG) technique in the VR literature. As shown in Alg. 1, the total number of $T$ update iterations is divided into $S$ epochs, each consisting of $m$ iterations. At the end of each epoch $s + 1$, the algorithm stores a snapshot of the expected full gradient $\mathcal{J}(\widetilde{\boldsymbol{\theta}}^s; \widetilde{\boldsymbol{\theta}}^s)$ (Line 4). Then, during each update iteration $k$ of epoch $s$, the

---

**Algorithm 1** Proposed Algorithm: Stochastic Performative Prediction with Variance Reduction (SPRINT)

**Input:** Number of total rounds $T$, number of iterations $m$ each epoch, learning rates $\gamma_1, \cdots, \gamma_m$, initialization $\widetilde{\boldsymbol{\theta}}^0 = \boldsymbol{\theta}_m^0 = \boldsymbol{\theta}_0$.
1: **for** $t = 0$ to $T - 1$ **do**
2: $\quad S = \lceil T/m \rceil$
3: $\quad$ **for** $s = 0$ to $S - 1$ **do**
4: $\quad\quad \boldsymbol{\theta}_0^{s+1} = \boldsymbol{\theta}_m^s = \widetilde{\boldsymbol{\theta}}^s$
5: $\quad\quad \nabla\mathcal{J}(\widetilde{\boldsymbol{\theta}}^s; \widetilde{\boldsymbol{\theta}}^s) = \frac{1}{n}\sum_{i=1}^n [\nabla\ell(z_i; \widetilde{\boldsymbol{\theta}}^s)]$ where each $z_i \sim \mathcal{D}(\widetilde{\boldsymbol{\theta}}^s)$
6: $\quad\quad$ **for** $k = 0$ to $m - 1$ **do**
7: $\quad\quad\quad$ Pick a sample $z_{i_k}$ where $z_{i_k} \sim \mathcal{D}(\boldsymbol{\theta}_k^{s+1})$
8: $\quad\quad\quad \boldsymbol{v}_k^{s+1} = \nabla\ell(z_{i_k}; \boldsymbol{\theta}_k^{s+1}) - \nabla\ell(z_{i_k}; \widetilde{\boldsymbol{\theta}}^s) + \nabla\mathcal{J}(\widetilde{\boldsymbol{\theta}}^s; \widetilde{\boldsymbol{\theta}}^s)$
9: $\quad\quad\quad \widetilde{\boldsymbol{\theta}}_{k+1}^{s+1} = \widetilde{\boldsymbol{\theta}}_k^{s+1} - \gamma_k \boldsymbol{v}_k^{s+1}$
10: $\quad\quad$ **end for**
11: $\quad\quad \widetilde{\boldsymbol{\theta}}^{s+1} = \boldsymbol{\theta}_m^{s+1}$
12: $\quad$ **end for**
13: **end for**
**Output:** $\boldsymbol{\theta}_m^S$

---

variance reduction step computes $\boldsymbol{v}_k^{s+1}$ (Line 8), where the variance of the current stochastic gradient estimate $\nabla\ell(z_{ik}; \boldsymbol{\theta}_k^{s+1})$ is offset by the variance of the previous stochastic gradient estimate $\nabla\ell(z_{ik}; \widetilde{\boldsymbol{\theta}}^s)$. Meanwhile, this step adds the expected gradient $\nabla\mathcal{J}(\widetilde{\boldsymbol{\theta}}^s; \widetilde{\boldsymbol{\theta}}^s)$ back to ensure that the overall expectation remains stable.

It is important to note that, unlike variance reduction in the non-PP settings, $\boldsymbol{v}_k^s$ is *not* an unbiased estimator of $\nabla\mathcal{J}(\boldsymbol{\theta}_k^s; \boldsymbol{\theta}_k^s) = \mathbb{E}_{z \sim \mathcal{D}(\boldsymbol{\theta}_k^s)}[\nabla\ell(z; \boldsymbol{\theta}_k^s)]$ in the PP setting. This is because $\mathcal{D}(\widetilde{\boldsymbol{\theta}}^{s-1}) \neq \mathcal{D}(\boldsymbol{\theta}_k^s)$. This introduces **new bias** compared to the non-PP setting and significantly complicates the convergence analysis of the algorithm, as we detail in Sec. 5. Table 1 summarizes a comparison of our algorithm with existing SGD-GD-Based PP methods. Notably, only Li & Wai (2024) and our method address the non-convex PP setting. However, our SPRINT approach achieves both a *faster* convergence rate and a (non-vanishing) error neighborhood that is *independent* of the variance of the stochastic gradient estimate, as illustrated in the next section.

## 5 CONVERGENCE ANALYSIS OF OUR SPRINT ALGORITHM

In this section, we analyze the convergence of the proposed SPRINT algorithm.

**1) Assumptions:** Our theoretical analysis only relies on the standard smoothness of the loss function (Assumption 3.1), the $\epsilon$-sensitivity of the distribution map (Def. 3.2), and the Lipschitzness of loss function (Assumption 3.3). Most notably, unlike Li & Wai (2024), we **do not** assume bounded variance of the stochastic gradient estimate (Assumption 3.4).

**2) Overview of the Proofs of the Main Theoretical Results:** Motivated by the analysis in (Reddi et al., 2016), which solely focused on SVRG in the non-PP settings, we aim to establish the convergence of SPRINT by constructing a suitable *Lyapunov function* and showing that the corresponding function sequence decreases over iterations. However, it turns out that one *cannot* trivially extend (Reddi et al., 2016) because the loss $\ell(\boldsymbol{\theta}; z)$ in the non-PP setting solely depends on the current model parameter $\boldsymbol{\theta}$. In stark contrast, the decoupled performative risk $\mathcal{J}(\boldsymbol{\theta}; \boldsymbol{\theta}') = \mathbb{E}_{z \sim \mathcal{D}(\boldsymbol{\theta}')}[\ell(\boldsymbol{\theta}; z)]$

in the PP setting depends on *two variables*: 1) $\boldsymbol{\theta}$, which determines the model prediction, and 2) $\boldsymbol{\theta}'$, which controls the data distribution. Because updates to the model parameter $\boldsymbol{\theta}$ simultaneously affect the data distribution, bounding the difference between consecutive terms in the Lyapunov sequence becomes significantly more challenging, which necessitates new proof and analysis techniques.

In what follows, we overcome these above challenges and establish the SPS convergence of SPRINT in three steps: (i) We construct an intermediate function sequence (i.e., $R_k^{s+1}(\boldsymbol{\theta})$ in Lemma 5.1), where consecutive terms differ only in the second argument of decoupled performative risk $\mathcal{J}$ and show that $R_k^{s+1}(\boldsymbol{\theta}_k^{s+1})$ is smaller than $R_{k+1}^{s+1}(\boldsymbol{\theta}_k^{s+1})$; (ii) We define the final Lyapunov function ($\widetilde{R}_k^{s+1}$ in Lemma 5.3) where consecutive terms differ in both arguments of $\mathcal{J}$; (iii) by proving the decreasing nature of $\widetilde{R}_k^{s+1}$, we establish the convergence of Algorithm 1 in Theorem 5.4 and derive the IFO complexity bound in Corollary 5.5.

**Step 1) Construct an intermediate function sequence:** In the $s+1$st epoch, we identify an intermediate function evaluated at the iterations $k$ and $k+1$ as:

$$R_k^{s+1}(\boldsymbol{\theta}) \triangleq \mathbb{E}\left[\mathcal{J}(\boldsymbol{\theta}_k^{s+1};\boldsymbol{\theta}) + c_k\|\boldsymbol{\theta}_k^{s+1} - \widetilde{\boldsymbol{\theta}}^s\|^2\right],$$

$$R_{k+1}^{s+1}(\boldsymbol{\theta}) \triangleq \mathbb{E}\left[\mathcal{J}(\boldsymbol{\theta}_{k+1}^{s+1};\boldsymbol{\theta}) + c_{k+1}\|\boldsymbol{\theta}_{k+1}^{s+1} - \widetilde{\boldsymbol{\theta}}^s\|^2\right].$$

Then, the following Lemma 5.1 characterizes the relation between $R_k^{s+1}(\boldsymbol{\theta}_k^{s+1})$ and $R_{k+1}^{s+1}(\boldsymbol{\theta}_k^{s+1})$:

**Lemma 5.1.** *Let $\beta_k$ be some positive constant, and $c_k, c_{k+1}$ are some constants defined in intermediate functions $R_k^{s+1}, R_{k+1}^{s+1}$ that satisfy the following:*

$$c_k - \frac{1}{2}\gamma_k = c_{k+1}(1 + \gamma_k\beta_k + 2L_0\epsilon\gamma_k) + (2L^2 + L_0^2\epsilon^2)(L\gamma_k^2 + 2c_{k+1}\gamma_k^2) + \frac{\beta_k}{2}L_0\epsilon\gamma_k$$

*Then, we have*

$$R_{k+1}^{s+1}(\boldsymbol{\theta}_k^{s+1}) \leq R_k^{s+1}(\boldsymbol{\theta}_k^{s+1}) - \Gamma_k \cdot \mathbb{E}[\|\nabla\mathcal{J}(\boldsymbol{\theta}_k^{s+1};\boldsymbol{\theta}_k^{s+1})\|^2] - \frac{1}{2}\gamma_k\|\boldsymbol{\theta}_k^{s+1} - \widetilde{\boldsymbol{\theta}}^s\|^2,$$

*where $\Gamma_k = \left(\gamma_k - \frac{c_{k+1}\gamma_k + \frac{1}{2}L_0\epsilon\gamma_k}{\beta_k} - 2L\gamma_k^2 - 4c_{k+1}\gamma_k^2\right)$.*

Note that $R_{k+1}^{s+1}(\boldsymbol{\theta}_k^{s+1})$ includes a term $\mathbb{E}[\mathcal{J}(\boldsymbol{\theta}_{k+1}^{s+1};\boldsymbol{\theta}_k^{s+1})]$, which represents the expected performative risk when the model parameter is $\boldsymbol{\theta}_{k+1}^{s+1}$ but the data distribution is induced by $\boldsymbol{\theta}_k^{s+1}$. This enables us to focus on bounding the influence of $\boldsymbol{\theta}$ through the first argument of $\mathcal{J}$, similar to analyses in the non-PP settings. However, since the snapshot $\mathcal{J}(\widetilde{\boldsymbol{\theta}}^s;\widetilde{\boldsymbol{\theta}}^s)$ corresponds to the full gradient computed under a *previous* distribution, we still need to bound the influence of distribution shifts. This implies that we need to bound the influence of distribution shifts from $\mathcal{D}(\widetilde{\boldsymbol{\theta}}^s)$ to $\mathcal{D}(\boldsymbol{\theta}_k^{s+1})$. To this end, we provide the following auxiliary lemma.

**Lemma 5.2** (Li & Wai (2024)). *Under Assumption 3.3 and $\epsilon$-sensitivity in Def. 3.2, we have:*

$$|\mathcal{J}(\boldsymbol{\theta};\boldsymbol{\theta}_1) - \mathcal{J}(\boldsymbol{\theta};\boldsymbol{\theta}_2)| \leq L_0\epsilon\|\boldsymbol{\theta}_1 - \boldsymbol{\theta}_2\|, \quad \forall\boldsymbol{\theta}.$$

We refer readers to (Li & Wai, 2024, App. B) for the proof of Lemma 5.2 which relies on $\epsilon$-sensitivity and Assumption 3.3.

To prove Lemma 5.1, we first apply the smoothness conditions of $\ell$ on $\mathbb{E}[\mathcal{J}(\boldsymbol{\theta}_{k+1}^{s+1};\boldsymbol{\theta}_k^{s+1})]$ and leverage Lemma 5.2 to bound $\|\widetilde{\boldsymbol{v}}_k^{s+1}\|$ and $\|\boldsymbol{v}_k^{s+1}\|^2$. The complete proof of Lemma 5.1 is provided in App. B due to space limitation.

**Step 2) Construct the final Lyapunov function:** We then construct the final Lyapunov function: at epoch $s+1$, we define Lyapunov function evaluated at iterations $k$ and $k+1$ as:

$$\widetilde{R}_k^{s+1} \triangleq \mathbb{E}[\mathcal{J}(\boldsymbol{\theta}_k^{s+1};\boldsymbol{\theta}_k^{s+1})] + c_k\|\boldsymbol{\theta}_k^{s+1} - \widetilde{\boldsymbol{\theta}}^s\|^2,$$

$$\widetilde{R}_{k+1}^{s+1} \triangleq \mathbb{E}[\mathcal{J}(\boldsymbol{\theta}_{k+1}^{s+1};\boldsymbol{\theta}_{k+1}^{s+1})] + c_{k+1}\|\boldsymbol{\theta}_{k+1}^{s+1} - \widetilde{\boldsymbol{\theta}}^s\|^2.$$

The following Lemma 5.3 characterizes the relation between $\widetilde{R}_{k+1}^s$ and $\widetilde{R}_{k+1}^{s+1}$:

**Lemma 5.3.** *Let $c_k, c_{k+1}, \beta_k, \Gamma_k$ be defined the same as in Lemma 5.1, we have:*

$$\widetilde{R}_{k+1}^{s+1} \leq \widetilde{R}_k^{s+1} + (L_0^4\epsilon^4 + 2L_0^2\epsilon^2 L^2 + 2L_0^2\epsilon^2)\gamma_k - \left(\Gamma_k - \frac{\gamma_k}{4}\right) \cdot \mathbb{E}\left[\|\nabla\mathcal{J}(\boldsymbol{\theta}_k^{s+1}; \boldsymbol{\theta}_k^{s+1})\|^2\right]. \quad (6)$$

Lemma 5.3 can be proved by leveraging Lemmas 5.1 and 5.2. The proof also establishes a bound on the difference between $\mathcal{J}(\boldsymbol{\theta}_k^{s+1}; \boldsymbol{\theta}_k^{s+1})$ and $\mathcal{J}(\boldsymbol{\theta}_{k+1}^{s+1}; \boldsymbol{\theta}_{k+1}^{s+1})$. The full proof is provided in App. C.

**Step 3) Establish the convergence of the** SPRINT **algorithm:** Building on the results of Lemma 5.3, we are now ready to prove the main convergence theorem:

**Theorem 5.4** (Convergence to an SPS point). *Consider* SPRINT *in Algorithm 1 with $T$ rounds and $S = \lceil T/m \rceil$ epochs, where each epoch consists of $m$ iterations. For the final iteration of each epoch, let the constant $c_m$ in the Lyapunov function be set to 0. Then we have:*

$$\frac{1}{T}\sum_{s=0}^{S-1}\sum_{k=0}^{m-1}\mathbb{E}\|\nabla\mathcal{J}(\boldsymbol{\theta}_k^{s+1}; \boldsymbol{\theta}_k^{s+1})\|^2 \leq \frac{\Delta_0}{T\Gamma} + \Delta_1, \quad (7)$$

*where terms $\Delta_0$ and $\Delta_1$ are defined as*

$$\Delta_0 \triangleq \mathcal{J}(\boldsymbol{\theta}_0^0; \boldsymbol{\theta}_0^0) - \ell_*; \quad \Delta_1 \triangleq \frac{(L_0^4\epsilon^4 + 2L_0^2\epsilon^2 L^2 + 2L_0^2\epsilon^2)\sum_{s=0}^{S}\sum_{k=0}^{m}\gamma_k}{T\Gamma}$$

*and $\Gamma > 0$ is a lower bound of $\Gamma_k - \frac{\gamma_k}{4}$ and is guaranteed to exist as a constant. Thus, Algorithm 1 achieves a convergence rate of $\mathcal{O}(1/T)$ and an variance-independent error neighborhood $\Delta_1$ of $\mathcal{O}(\epsilon^2 + \epsilon^4)$.*

Eqn. 7 is established by summing the terms in Lemma 5.3, and then lower bounding $\Gamma_k - \frac{1}{4}\gamma_k$ to guarantee the convergence rate $\mathcal{O}(1/T)$. The complete proof can be found in App. D. Thm. 5.4 improves upon previous results in three key aspects: (i) the $\mathcal{O}(1/T)$ convergence rate is faster than that of Li & Wai (2024); (ii) we eliminate Assumption 3.4, and the error neighborhood $\Delta_1$ is **no longer dependent** on the variance of the stochastic gradient; and (iii) the error neighborhood is $\mathcal{O}(\epsilon^2 + \epsilon^4)$, which is guaranteed to be smaller than that of the SGD-GD-Based PP algorithm in (Li & Wai, 2024) when the distribution map is not sensitive ($\epsilon < 1$). Next, we derive the IFO complexity as defined in Reddi et al. (2016) (i.e., the number of operations required to sample and compute the gradient to achieve a $\mathcal{O}(\delta)$-SPS) for Algorithm 1. Compared to the IFO Complexity of SGD-GD ($\mathcal{O}(\frac{(\Delta_0 + \frac{L}{2}\sigma_0^2)^2}{\delta^2})$), SPRINT has a clear advantage when the population size $n$ is not too large or when the variance parameter $\sigma_0^2$ is large or even does not exist. We provide more details in App. H.

**Corollary 5.5** (IFO Complexity of SPRINT). *Assume a finite distribution setting where the population consists of $n$ samples, and $\alpha \in (0, 1)$. For the algorithm with fixed parameters at each round, i.e., $\gamma_k = \gamma = \mathcal{O}(\frac{1}{n^\alpha}), \beta_k = \beta = \mathcal{O}(n^{\frac{\alpha}{2}}), m = \mathcal{O}(n^{\frac{\alpha}{2}})$, we have $\Gamma = \mathcal{O}(\frac{1}{n^\alpha})$, and the IFO complexity to achieve $\mathcal{O}(\delta)$-SPS, in addition to the error neighborhood, is $\mathcal{O}(\frac{(n^\alpha + n^{1+\frac{\alpha}{2}})\Delta_0}{\delta})$.*

# 6 EXPERIMENTS

In this section, we conduct comprehensive experimental studies to verify the effectiveness of SPRINT in practice. We compare the convergence behavior of the training loss and training accuracy of SPRINT with SGD-GD (Li & Wai, 2024) on 3 real-world datasets with non-convex models and study the performative effects. The results of Credit dataset (Kaggle, 2012) and CIFAR-10 dataset (Krizhevsky et al., 2009) are included in the main paper, while results of MNIST dataset (Deng, 2012) are relegated in App. F due to space limitation. We also show the magnitude of the squared gradient norm $\|\nabla\mathcal{J}(\widetilde{\boldsymbol{\theta}}^s; \widetilde{\boldsymbol{\theta}}^s)\|^2$ in App. F to validate Thm. 5.4.

Besides the credit and MNIST datasets, we are the first to provide results using CNN models on the CIFAR-10 (Krizhevsky et al., 2009) dataset in the PP settings. As neural network architectures are widely used, our experiments go one step further for the practical applications of PP algorithms.

## 6.1 EXPERIMENTS ON THE CREDIT DATASET

We use the *Give me some credit* data (Kaggle, 2012) with 10 features $X \in \mathbb{R}^{10}$ to predict creditworthiness $Y \in \{0, 1\}$ (Perdomo et al., 2021; Jin et al., 2024). We preprocess the dataset by selecting a

balanced subset of $5,000$ examples and then normalizing features similar to Perdomo et al. (2021). We then assume a credit scoring agency uses a two-layer MLP model to predict $Y$ given the individuals' features $X$ with cross entropy loss. To simulate performative effects, we assume individuals are strategic (Perdomo et al., 2021; Hardt et al., 2015), i.e., there is a subset of features $X_s \in X$ which individuals can change to $X'_s$ based on the current model. Given the current model $f_{\boldsymbol{\theta}}$, individuals with feature $x_s$ will best respond to the model in the direction of $\nabla_x f_{\boldsymbol{\theta}}(x_s)$ and change $x_s$ to $x'_s = x_s + \alpha \nabla_x f_{\boldsymbol{\theta}}(x_s)$ to improve their scoring (Rosenfeld et al., 2020; Xie & Zhang, 2024).

In this setting, we train the MLP model using SPRINT and SGD-GD for 40 epochs and illustrate the accuracy and training loss in Fig. 1. We simulate different performative effects using of $\alpha \in \{0.01, 0.2, 0.4\}$. In Fig. 1, we show that the SPRINT algorithm achieves lower training loss and converges faster compared with the SGD-GD in all 3 cases and the advantages are quite prominent when performative effects are larger (i.e., when $\alpha = 0.2$ and $0.4$).

## 6.2 Experiments on the CIFAR-10 dataset

Furthermore, we use the **CIFAR-10 dataset (Krizhevsky et al., 2009)** to predict 10 possible class labels $Y \in \{0, 1, ..., 9\}$. Each label corresponds to a common and well-separated category such as *dog* and *truck*. The dataset contains $50,000$ images in total and each class includes $5,000$ samples. We design a CNN model with two convolutional layers to train the algorithms with a $0.05$ learning rate and cross-entropy loss. We simulate performative effects motivated by retention dynamics (Hashimoto et al., 2018; Jin et al., 2024; Zhang et al., 2019) where the class distribution will change based on the performance of the current model. The fraction in class $c$ at iteration $k$ of $p_c^{k+1} = \frac{e^{-\alpha \ell_c^k}}{\sum_{c'} e^{-\alpha \ell_{c'}^k}}$, where $\ell_c^k$ is the loss expectation of class $c$ in iteration $k$. This means that the image class with larger loss now will be less likely to appear in the next iteration (Zhang et al., 2019). We set $\alpha$ at $20, 50, 80$ and train 400 epochs in total. The experimental results are shown in Fig. 2. Consistent with the results observed on the Credit dataset, SPRINT also exhibits a faster convergence on the CIFAR-10 dataset.

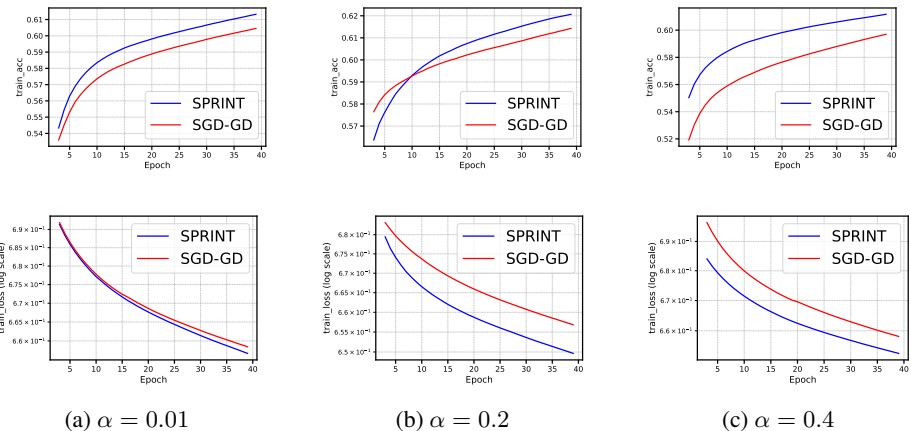

(a) $\alpha = 0.01$          (b) $\alpha = 0.2$          (c) $\alpha = 0.4$

Figure 1: The **Credit** dataset: training accuracy (first row) and training loss (second row) of SGD-GD and SPRINT under different $\alpha$. A larger $\alpha$ implies more intense individual strategic behaviors.

## 7 Conclusion and future work

This paper focused on improving the convergence of stochastic optimization methods in non-convex PP settings by adapting a variance reduction method. We proposed SPRINT as a new algorithm to achieve better convergence results without the need for the bounded variance assumption on stochastic gradient descent estimates. We also presented extensive experiments with different non-convex neural network models to compare SPRINT with SGD-GD, which demonstrated the superior performance of our algorithm SPRINT. However, SPRINT only converges to a non-vanishing error neighborhood. Future work may include developing algorithms to converge to performative optimal

solutions in the non-convex PP settings, or adapting other more advanced variance reduction methods (SARAH or SPIDER) to the PP setting and trying to completely eliminate the non-vanishing error neighborhood.

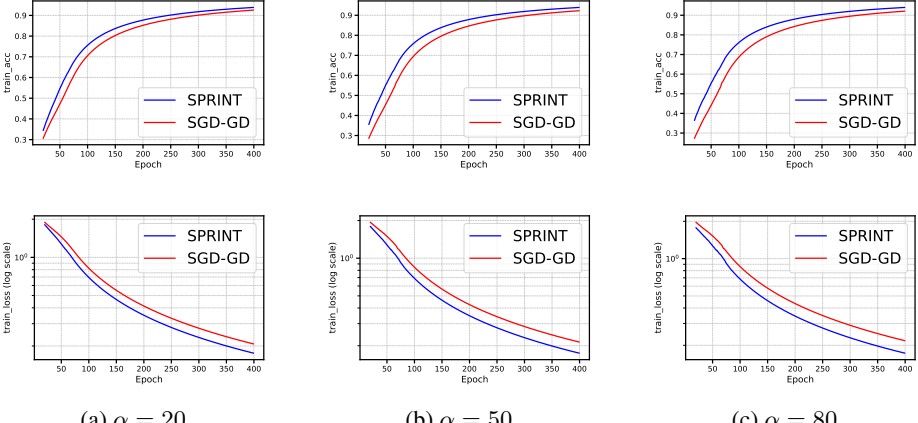

(a) $\alpha = 20$       (b) $\alpha = 50$       (c) $\alpha = 80$

Figure 2: The **CIFAR-10** dataset: training accuracy (first row) and training loss (second row) of SGD-GD and SPRINT under different $\alpha$. A larger $\alpha$ implies more intense retention dynamics.

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

## A   THE USE OF LARGE LANGUAGE MODELS (LLMS)

We only use LLMs for polishing the writing without using them to generate any single sentence solely on their own.

## B   PROOF OF LEMMA 5.1

*Lemma.* Let $\beta_k$ be some positive constant, and $c_k, c_{k+1}$ are some constants defined in intermediate functions $R_k^{s+1}, R_{k+1}^{s+1}$ that satisfy the following

$$c_k - \frac{1}{2}\gamma_k = c_{k+1}(1 + \gamma_k\beta_k + 2L_0\epsilon\gamma_k) + (2L^2 + L_0^2\epsilon^2)(L\gamma_k^2 + 2c_{k+1}\gamma_k^2) + \frac{\beta_k}{2}L_0\epsilon\gamma_k$$

Then, we have

$$R_{k+1}^{s+1}(\boldsymbol{\theta}_k^{s+1}) \le R_k^{s+1}(\boldsymbol{\theta}_k^{s+1}) - \Gamma_k \cdot \mathbb{E}[\|\nabla\mathcal{J}(\boldsymbol{\theta}_k^{s+1};\boldsymbol{\theta}_k^{s+1})\|^2] - \frac{1}{2}\gamma_k\|\boldsymbol{\theta}_k^{s+1} - \widetilde{\boldsymbol{\theta}}^s\|^2$$

where $\Gamma_k = \left(\gamma_k - \frac{c_{k+1}\gamma_k + \frac{1}{2}L_0\epsilon\gamma_k}{\beta_k} - 2L\gamma_k^2 - 4c_{k+1}\gamma_k^2\right)$

*Proof.* This proof mainly relies on Assumptions 3.1, 3.3, and $\epsilon$-sensitivity.

Since $\ell(\boldsymbol{\theta};z)$ is $L$-smooth in $\boldsymbol{\theta}$ and $v_k^{s+1} = \frac{1}{\gamma_k}\left(\boldsymbol{\theta}_k^{s+1} - \boldsymbol{\theta}_{k+1}^{s+1}\right)$, we can apply descent lemma to get the following result:

$$\mathbb{E}[\mathcal{J}(\boldsymbol{\theta}_{k+1}^{s+1};\boldsymbol{\theta}_k^{s+1})] \le \mathbb{E}[\mathcal{J}(\boldsymbol{\theta}_k^{s+1};\boldsymbol{\theta}_k^{s+1}) + \frac{L}{2}\|\boldsymbol{\theta}_{k+1}^{s+1} - \boldsymbol{\theta}_k^{s+1}\|^2 + \nabla\mathcal{J}(\boldsymbol{\theta}_k^{s+1};\boldsymbol{\theta}_k^{s+1})^T(\boldsymbol{\theta}_{k+1}^{s+1} - \boldsymbol{\theta}_k^{s+1})] \quad (8)$$

$$= \mathbb{E}[\mathcal{J}(\boldsymbol{\theta}_k^{s+1};\boldsymbol{\theta}_k^{s+1})] + \frac{L\gamma_k^2}{2}\|v_k^{s+1}\|^2 - \gamma_k\nabla\mathcal{J}(\boldsymbol{\theta}_k^{s+1};\boldsymbol{\theta}_k^{s+1})^T v_k^{s+1}$$

By definition of $v_k^{s+1}$, we have:

$$\mathbb{E}[v_k^{s+1}] = \mathbb{E}[\nabla\ell(z_{i_k};\boldsymbol{\theta}_k^{s+1}) - \nabla\ell(z_{i_k};\widetilde{\boldsymbol{\theta}}^s) + \nabla\mathcal{J}(\widetilde{\boldsymbol{\theta}}^s;\widetilde{\boldsymbol{\theta}}^s)] \quad (9)$$

$$= \mathbb{E}[\nabla\ell(z_{i_k};\boldsymbol{\theta}_k^{s+1}) - \nabla\ell(z_{i_k};\widetilde{\boldsymbol{\theta}}^s) + \nabla\mathcal{J}(\widetilde{\boldsymbol{\theta}}^s;\boldsymbol{\theta}_k^{s+1})] - \mathbb{E}[\nabla\mathcal{J}(\widetilde{\boldsymbol{\theta}}^s;\boldsymbol{\theta}_k^{s+1}) - \nabla\mathcal{J}(\widetilde{\boldsymbol{\theta}}^s;\widetilde{\boldsymbol{\theta}}^s)]$$

$$= \mathbb{E}[\nabla\mathcal{J}(\boldsymbol{\theta}_k^{s+1};\boldsymbol{\theta}_k^{s+1})] - \mathbb{E}[\nabla\mathcal{J}(\widetilde{\boldsymbol{\theta}}^s;\boldsymbol{\theta}_k^{s+1}) - \nabla\mathcal{J}(\widetilde{\boldsymbol{\theta}}^s;\widetilde{\boldsymbol{\theta}}^s)]$$

The last equality holds since $\mathbb{E}[\nabla\ell(z_{i_k};\widetilde{\boldsymbol{\theta}}^s)] = \nabla\mathcal{J}(\widetilde{\boldsymbol{\theta}}^s,\boldsymbol{\theta}_k^{s+1})$. Then applying Lemma 5.2 on the second term above to derive the following:

$$\gamma_k\nabla\mathcal{J}(\boldsymbol{\theta}_k^{s+1};\boldsymbol{\theta}_k^{s+1})^T v_k^{s+1} = \gamma_k\nabla\mathcal{J}(\boldsymbol{\theta}_k^{s+1};\boldsymbol{\theta}_k^{s+1})^T\left(\nabla\mathcal{J}(\boldsymbol{\theta}_k^{s+1};\boldsymbol{\theta}_k^{s+1}) - \mathbb{E}[\nabla\mathcal{J}(\widetilde{\boldsymbol{\theta}}^s;\boldsymbol{\theta}_k^{s+1}) - \nabla\mathcal{J}(\widetilde{\boldsymbol{\theta}}^s;\widetilde{\boldsymbol{\theta}}^s)]\right)$$

$$\ge \gamma_k\|\nabla\mathcal{J}(\boldsymbol{\theta}_k^{s+1};\boldsymbol{\theta}_k^{s+1})\|^2 - L_0\epsilon\gamma_k\|\nabla\mathcal{J}(\boldsymbol{\theta}_k^{s+1};\boldsymbol{\theta}_k^{s+1})\|\|\widetilde{\boldsymbol{\theta}}^s - \boldsymbol{\theta}_k^{s+1}\|$$

Next, applying Young's Inequality, we can get:

$$8 \le \mathbb{E}[\mathcal{J}(\boldsymbol{\theta}_{k+1}^{s+1};\boldsymbol{\theta}_k^{s+1}) - \gamma_k\|\nabla\mathcal{J}(\boldsymbol{\theta}_k^{s+1};\boldsymbol{\theta}_k^{s+1})\|^2 + L_0\epsilon\gamma_k(\frac{1}{2\beta_k}\|\nabla\mathcal{J}(\boldsymbol{\theta}_k^{s+1};\boldsymbol{\theta}_k^{s+1})\|^2 \quad (10)$$

$$+ \frac{\beta_k}{2}\|\boldsymbol{\theta}_k^{s+1} - \widetilde{\boldsymbol{\theta}}^s\|^2) + \frac{L\gamma_k^2}{2}\|v_k^{s+1}\|^2]$$

Next, bound $\mathbb{E}[\|\boldsymbol{\theta}_{k+1}^{s+1} - \widetilde{\boldsymbol{\theta}}^s\|^2]$ by adding and subtracting as follows:

$$\mathbb{E}[\|\boldsymbol{\theta}_{k+1}^{s+1} - \widetilde{\boldsymbol{\theta}}^s\|^2] = \mathbb{E}[\|\boldsymbol{\theta}_{k+1}^{s+1} - \boldsymbol{\theta}_k^{s+1} + \boldsymbol{\theta}_k^{s+1} - \widetilde{\boldsymbol{\theta}}^s\|^2]$$

$$= \mathbb{E}[\gamma_k^2 \|v_k^{s+1}\|^2 + \|\boldsymbol{\theta}_k^{s+1} - \widetilde{\boldsymbol{\theta}}^s\|^2] + 2\gamma_k \mathbb{E}[\|\boldsymbol{\theta}_k^{s+1} - \widetilde{\boldsymbol{\theta}}^s\| \|v_k^{s+1}\|]$$

$$\leq \mathbb{E}[\gamma_k^2 \|v_k^{s+1}\|^2 + \|\boldsymbol{\theta}_k^{s+1} - \widetilde{\boldsymbol{\theta}}^s\|^2] + 2\gamma_k \mathbb{E}[\frac{1}{2\beta_k} \|\nabla \mathcal{J}(\boldsymbol{\theta}_k^{s+1}; \boldsymbol{\theta}_k^{s+1})\|^2 \qquad (11)$$

$$+ \frac{\beta_k}{2} \|\boldsymbol{\theta}_k^{s+1} - \widetilde{\boldsymbol{\theta}}^s\|^2] + 2L_0\epsilon\gamma_k \mathbb{E}\|\widetilde{\boldsymbol{\theta}}^s - \boldsymbol{\theta}_k^{s+1}\|^2$$

Similary, $\mathbb{E}[\|v_k^{s+1}\|^2]$ can also be bounded:

$$\mathbb{E}[\|v_k^{s+1}\|^2] = \mathbb{E}\left[\nabla \ell(z_{i_k}; \boldsymbol{\theta}_k^{s+1}) - \nabla \ell(z_{i_k}; \widetilde{\boldsymbol{\theta}}^s) + \nabla \mathcal{J}(\widetilde{\boldsymbol{\theta}}^s; \widetilde{\boldsymbol{\theta}}^s)\right]^2 \qquad (12)$$

$$\leq 2\mathbb{E}\left[\|\nabla \ell(z_{i_k}; \boldsymbol{\theta}_k^{s+1}) - \nabla \ell(z_{i_k}; \widetilde{\boldsymbol{\theta}}^s) + \nabla \mathcal{J}(\widetilde{\boldsymbol{\theta}}^s; \boldsymbol{\theta}_k^{s+1})\|^2\right]$$

$$+ 2\mathbb{E}\left[\|\nabla \mathcal{J}(\widetilde{\boldsymbol{\theta}}^s; \boldsymbol{\theta}_k^{s+1}) - \nabla \mathcal{J}(\widetilde{\boldsymbol{\theta}}^s; \widetilde{\boldsymbol{\theta}}^s)\|^2\right]$$

The above inequality is obtained by adding and subtracting $\nabla \mathcal{J}(\widetilde{\boldsymbol{\theta}}^s; \boldsymbol{\theta}_k^{s+1})$ and then applying Young's Inequality. For the first term, it is worth noting that the term already gets rid of the performative effects, i.e., the distribution is fixed as $\mathcal{D}(\boldsymbol{\theta}_k^{s+1})$. Then bounding this term is equivalent to bound the norm of update in Reddi et al. (2016). Specifically, let $\delta_k^{s+1} = \nabla \ell(z_{i_k}; \boldsymbol{\theta}_k^{s+1}) - \nabla \ell(z_{i_k}; \widetilde{\boldsymbol{\theta}}^s)$, we know $\mathbb{E}[\delta_k^{s+1}] = \mathbb{E}[\ell(z_{i_k}; \boldsymbol{\theta}_k^{s+1})] - \mathbb{E}[\nabla \ell(z_{i_k}; \widetilde{\boldsymbol{\theta}}^s)] = \nabla \mathcal{J}(\boldsymbol{\theta}_k^{s+1}; \boldsymbol{\theta}_k^{s+1}) - \nabla \mathcal{J}(\widetilde{\boldsymbol{\theta}}^s; \boldsymbol{\theta}_k^{s+1})$. Thus, we have:

$$\mathbb{E}\left[\|\nabla \ell(z_{i_k}; \boldsymbol{\theta}_k^{s+1}) - \nabla \ell(z_{i_k}; \widetilde{\boldsymbol{\theta}}^s) + \nabla \mathcal{J}(\widetilde{\boldsymbol{\theta}}^s; \boldsymbol{\theta}_k^{s+1})\|^2\right] \qquad (13)$$

$$= \mathbb{E}[\|\delta_k^{s+1} + \nabla \mathcal{J}(\widetilde{\boldsymbol{\theta}}^s; \boldsymbol{\theta}_k^{s+1})\|^2]$$

$$= \mathbb{E}[\|\delta_k^{s+1} + \nabla \mathcal{J}(\widetilde{\boldsymbol{\theta}}^s; \boldsymbol{\theta}_k^{s+1}) - \nabla \mathcal{J}(\boldsymbol{\theta}_k^{s+1}; \boldsymbol{\theta}_k^{s+1}) + \nabla \mathcal{J}(\boldsymbol{\theta}_k^{s+1}; \boldsymbol{\theta}_k^{s+1})\|^2]$$

$$= \mathbb{E}[\|\delta_k^{s+1} - \mathbb{E}[\delta_k^{s+1}] + \nabla \mathcal{J}(\boldsymbol{\theta}_k^{s+1}; \boldsymbol{\theta}_k^{s+1})\|^2]$$

$$\leq 2\mathbb{E}[\|\delta_k^{s+1} - \mathbb{E}[\delta_k^{s+1}]\|^2] + 2\mathbb{E}[\|\nabla \mathcal{J}(\boldsymbol{\theta}_k^{s+1}; \boldsymbol{\theta}_k^{s+1})\|^2]$$

$$\leq 2\mathbb{E}[\|\delta_k^{s+1}\|^2] + 2\mathbb{E}[\|\nabla \mathcal{J}(\boldsymbol{\theta}_k^{s+1}; \boldsymbol{\theta}_k^{s+1})\|^2]$$

$$\leq 2L^2 \mathbb{E}[\|\widetilde{\boldsymbol{\theta}}_k^{s+1} - \widetilde{\boldsymbol{\theta}}^s\|^2] + 2\mathbb{E}[\|\nabla \mathcal{J}(\boldsymbol{\theta}_k^{s+1}; \boldsymbol{\theta}_k^{s+1})\|^2]$$

Since $\nabla \mathcal{J}$ is $L_0$-lipschitz in $z$ (i.e., Assumption 3.1), according to the same reasoning in Lemma 5.2, we know $\|\nabla \mathcal{J}(\widetilde{\boldsymbol{\theta}}^s; \boldsymbol{\theta}_k^{s+1}) - \nabla \mathcal{J}(\widetilde{\boldsymbol{\theta}}^s; \widetilde{\boldsymbol{\theta}}^s)\| \leq L_0\epsilon\|\boldsymbol{\theta}_k^{s+1} - \widetilde{\boldsymbol{\theta}}^s\|$ always holds. So we have:

$$\mathbb{E}\left[\|\nabla \mathcal{J}(\widetilde{\boldsymbol{\theta}}^s; \boldsymbol{\theta}_k^{s+1}) - \nabla \mathcal{J}(\widetilde{\boldsymbol{\theta}}^s; \widetilde{\boldsymbol{\theta}}^s)\|^2\right] \leq L_0^2\epsilon^2 \mathbb{E}\|\boldsymbol{\theta}_k^{s+1} - \widetilde{\boldsymbol{\theta}}^s\|^2 \qquad (14)$$

Sum them together, we have:

$$\mathbb{E}[\|v_k^{s+1}\|^2] \leq 4\|\nabla \mathcal{J}(\boldsymbol{\theta}_k^{s+1}; \boldsymbol{\theta}_k^{s+1})\|^2 + (4L^2 + 2L_0^2\epsilon^2)\mathbb{E}\|\boldsymbol{\theta}_k^{s+1} - \widetilde{\boldsymbol{\theta}}^s\|^2 \qquad (15)$$

Now consider $R_{k+1}^{s+1}$ and plug 8 and 11 into its formula, we have:

$$R_{k+1}^{s+1}(\boldsymbol{\theta}_k^{s+1}) = \mathbb{E}\left[\mathcal{J}(\boldsymbol{\theta}_{k+1}^{s+1}; \boldsymbol{\theta}_k^{s+1}) + C_{k+1}\|\widetilde{\boldsymbol{\theta}}_{k+1}^{s+1} - \widetilde{\boldsymbol{\theta}}^s\|^2\right] \tag{16}$$

$$\leq \mathbb{E}[\mathcal{J}(\boldsymbol{\theta}_k^{s+1}; \boldsymbol{\theta}_k^{s+1}) - \gamma_k\|\nabla\mathcal{J}(\boldsymbol{\theta}_k^{s+1}, \boldsymbol{\theta}_k^{s+1})\|^2$$

$$+ L_0\epsilon\gamma_k(\frac{1}{2\beta_k}\|\nabla\mathcal{J}(\boldsymbol{\theta}_k^{s+1}; \boldsymbol{\theta}_k^{s+1})\|^2 + \frac{\beta_k}{2}\|\boldsymbol{\theta}_k^{s+1} - \widetilde{\boldsymbol{\theta}}^s\|^2)$$

$$+ \frac{L\gamma_k^2}{2}\|v_k^{s+1}\|^2] + \mathbb{E}[c_{k+1}\gamma_k^2\|v_k^{s+1}\|^2$$

$$+ c_{k+1}\|\boldsymbol{\theta}_{k+1}^{s+1} - \widetilde{\boldsymbol{\theta}}^s\|^2 + 2c_{k+1}\gamma_k\mathbb{E}(\frac{1}{2\beta_k}\|\nabla\mathcal{J}(\boldsymbol{\theta}_k^{s+1}; \boldsymbol{\theta}_k^{s+1})\|^2 \tag{17}$$

$$+ \frac{\beta_k}{2}\|\boldsymbol{\theta}_k^{s+1} - \widetilde{\boldsymbol{\theta}}^s\|^2)] + 2c_{k+1}L_0\epsilon\gamma_k\mathbb{E}\left\|\widetilde{\boldsymbol{\theta}}^s - \boldsymbol{\theta}_k^{s+1}\right\|^2$$

$$= \mathbb{E}\left[\mathcal{J}(\boldsymbol{\theta}_k^{s+1}; \boldsymbol{\theta}_k^{s+1})\right] + \left(\frac{L\gamma_k^2}{2} + c_{k+1}\gamma_k^2\right)\mathbb{E}\left[\|v_k^{s+1}\|^2\right]$$

$$- \left(\gamma_k - \frac{c_{k+1}\gamma_k + \frac{1}{2}L_0\epsilon\gamma_k}{\beta_k}\right)\mathbb{E}\left[\|\nabla\mathcal{J}(\boldsymbol{\theta}_k^{s+1}; \boldsymbol{\theta}_k^{s+1})\|^2\right]$$

$$+ \left(c_{k+1}(1 + \gamma_k\beta_k + 2L_0\epsilon\gamma_k) + \frac{\beta_kL_0\epsilon\gamma_k}{2}\right)\mathbb{E}\left[\|\widetilde{\boldsymbol{\theta}}^s - \boldsymbol{\theta}_k^{s+1}\|^2\right]$$

Regarding the $\|v_k^{s+1}\|^2$, we plug 15 in 16 to get the expression we need:

$$R_{k+1}^{s+1}(\boldsymbol{\theta}_k^{s+1}) \leq \mathbb{E}\left[\mathcal{J}(\boldsymbol{\theta}_k^{s+1}; \boldsymbol{\theta}_k^{s+1})\right] - \Gamma_k\mathbb{E}\left[\|\nabla\mathcal{J}(\boldsymbol{\theta}_k^{s+1}; \boldsymbol{\theta}_k^{s+1})\|^2\right] + (c_k - \frac{1}{2}\gamma_k)\mathbb{E}\|\boldsymbol{\theta}_k^{s+1} - \widetilde{\boldsymbol{\theta}}^s\|^2$$

$$= R_k^{s+1} - \Gamma_k\mathbb{E}\left[\|\nabla\mathcal{J}(\boldsymbol{\theta}_k^{s+1}; \boldsymbol{\theta}_k^{s+1})\|^2\right] - \frac{1}{2}\gamma_k\mathbb{E}\|\boldsymbol{\theta}_k^{s+1} - \widetilde{\boldsymbol{\theta}}^s\|^2$$

where

$$c_k - \frac{1}{2}\gamma_k = c_{k+1}(1 + \gamma_k\beta_k + 2L_0\epsilon\gamma_k) + (2L^2 + L_0^2\epsilon^2)(L\gamma_k^2 + 2c_{k+1}\gamma_k^2) + \frac{\beta_k}{2}L_0\epsilon\gamma_k$$

$$\Gamma_k = \left(\gamma_k - \frac{c_{k+1}\gamma_k + \frac{1}{2}L_0\epsilon\gamma_k}{\beta_k} - 2L\gamma_k^2 - 4c_{k+1}\gamma_k^2\right)$$

$$\square$$

## C  PROOF OF LEMMA 5.3

*Lemma.* With $c_k, c_{k+1}, \beta_k, \Gamma_k$ same as Lemma 5.1, denote

$$\widetilde{R}_k^{s+1} \triangleq \mathbb{E}[\mathcal{J}(\boldsymbol{\theta}_k^{s+1}; \boldsymbol{\theta}_k^{s+1}) + c_k\|\boldsymbol{\theta}_k^{s+1} - \widetilde{\boldsymbol{\theta}}^s\|^2]$$

$$\widetilde{R}_{k+1}^{s+1} \triangleq \mathbb{E}[\mathcal{J}(\boldsymbol{\theta}_{k+1}^{s+1}; \boldsymbol{\theta}_{k+1}^{s+1}) + c_{k+1}\|\boldsymbol{\theta}_{k+1}^{s+1} - \widetilde{\boldsymbol{\theta}}^s\|^2]$$

Then we have:

$$\widetilde{R}_{k+1}^{s+1} \leq \widetilde{R}_k^{s+1} + (L_0^4\epsilon^4 + 2L_0^2\epsilon^2L^2 + 2L_0^2\epsilon^2)\gamma_k - (\Gamma_k - \frac{\gamma_k}{4}) \cdot \mathbb{E}[\|\nabla\mathcal{J}(\boldsymbol{\theta}_k^{s+1}; \boldsymbol{\theta}_k^{s+1})\|^2] \tag{18}$$

*Proof.* This proof mainly relies on Assumptions 3.1, 3.3, and $\epsilon$-sensitivity. According to Lemma 5.1 (Eqn. 6), we have:

$$\mathbb{E}[\mathcal{J}(\boldsymbol{\theta}_{k+1}^{s+1}; \boldsymbol{\theta}_k^{s+1}) + c_{k+1}\|\boldsymbol{\theta}_{k+1}^{s+1} - \widetilde{\boldsymbol{\theta}}^s\|^2] \leq \mathbb{E}[\mathcal{J}(\boldsymbol{\theta}_k^{s+1}; \boldsymbol{\theta}_k^{s+1})] + c_k\|\boldsymbol{\theta}_k^{s+1} - \widetilde{\boldsymbol{\theta}}^s\|^2 \tag{19}$$

$$- \Gamma_k\|\nabla\mathcal{J}(\boldsymbol{\theta}_k^{s+1}; \boldsymbol{\theta}_k^{s+1})\|^2 - \frac{\gamma_k}{2}\|\boldsymbol{\theta}_k^{s+1} - \widetilde{\boldsymbol{\theta}}^s\|^2]$$

By aggregating terms, we will have:

$$c_{k+1}\mathbb{E}\|\boldsymbol{\theta}_{k+1}^{s+1} - \widetilde{\boldsymbol{\theta}}^s\|^2 - \left(c_k - \frac{\gamma_k}{2}\right)\mathbb{E}\|\boldsymbol{\theta}_k^{s+1} - \widetilde{\boldsymbol{\theta}}^s\|^2 + \Gamma_k\mathbb{E}\|\nabla\mathcal{J}(\boldsymbol{\theta}_k^{s+1};\boldsymbol{\theta}_k^{s+1})\|^2 \tag{20}$$
$$\leq \mathbb{E}[\mathcal{J}(\boldsymbol{\theta}_k^{s+1};\boldsymbol{\theta}_k^{s+1}) - \mathcal{J}(\boldsymbol{\theta}_{k+1}^{s+1};\boldsymbol{\theta}_k^{s+1})]$$

We can further bound the above expression by:

$$\begin{aligned}
\mathbb{E}[\mathcal{J}(\boldsymbol{\theta}_k^{s+1};\boldsymbol{\theta}_k^{s+1}) - \mathcal{J}(\boldsymbol{\theta}_{k+1}^{s+1};\boldsymbol{\theta}_k^{s+1})] &= \mathbb{E}\left[\mathcal{J}(\boldsymbol{\theta}_k^{s+1};\boldsymbol{\theta}_k^{s+1}) - \mathcal{J}(\boldsymbol{\theta}_{k+1}^{s+1};\boldsymbol{\theta}_{k+1}^{s+1})\right] \\
&\quad + \mathbb{E}\left[\mathcal{J}(\boldsymbol{\theta}_{k+1}^{s+1};\boldsymbol{\theta}_{k+1}^{s+1}) - \mathcal{J}(\boldsymbol{\theta}_{k+1}^{s+1};\boldsymbol{\theta}_k^{s+1})\right] \\
&\leq \mathbb{E}\left[\mathcal{J}(\boldsymbol{\theta}_k^{s+1};\boldsymbol{\theta}_k^{s+1}) - \mathcal{J}(\boldsymbol{\theta}_{k+1}^{s+1};\boldsymbol{\theta}_{k+1}^{s+1})\right] \\
&\quad + L_0\epsilon\mathbb{E}\|\boldsymbol{\theta}_{k+1}^{s+1} - \boldsymbol{\theta}_k^{s+1}\|
\end{aligned} \tag{21}$$

Regarding $\mathbb{E}\|\boldsymbol{\theta}_{k+1}^{s+1} - \boldsymbol{\theta}_k^{s+1}\| = \gamma_k\mathbb{E}\|v_k^{s+1}\|$, we can bound it according to Eqn. 9 and Eqn. 15:

$$\begin{aligned}
\mathbb{E}[\|v_k^{s+1}\|] &= \mathbb{E}[\|\nabla\ell(z_{i_k};\boldsymbol{\theta}_k^{s+1}) - \nabla\ell(z_{i_k};\widetilde{\boldsymbol{\theta}}^s) + \nabla\mathcal{J}(\widetilde{\boldsymbol{\theta}}^s;\widetilde{\boldsymbol{\theta}}^s)\|] \\
&\leq \mathbb{E}[\|\nabla\ell(z_{i_k};\boldsymbol{\theta}_k^{s+1}) - \nabla\ell(z_{i_k};\widetilde{\boldsymbol{\theta}}^s) + \nabla\mathcal{J}(\widetilde{\boldsymbol{\theta}}^s;\boldsymbol{\theta}_k^{s+1})\|] \\
&\quad + \mathbb{E}[\|\nabla\mathcal{J}(\widetilde{\boldsymbol{\theta}}^s;\boldsymbol{\theta}_k^{s+1}) - \nabla\mathcal{J}(\widetilde{\boldsymbol{\theta}}^s;\widetilde{\boldsymbol{\theta}}^s)\|]
\end{aligned} \tag{22}$$

Let $\nabla\ell(z_{i_k};\boldsymbol{\theta}_k^{s+1}) - \nabla\ell(z_{i_k};\widetilde{\boldsymbol{\theta}}^s) + \nabla\mathcal{J}(\widetilde{\boldsymbol{\theta}}^s;\boldsymbol{\theta}_k^{s+1})$ be $A$ and $\nabla\mathcal{J}(\widetilde{\boldsymbol{\theta}}^s;\boldsymbol{\theta}_k^{s+1}) - \nabla\mathcal{J}(\widetilde{\boldsymbol{\theta}}^s;\widetilde{\boldsymbol{\theta}}^s)$ be $B$. According to Eqn. 13, we have:

$$\mathbb{E}[\|A\|^2] \leq 2\mathbb{E}\left[\|\nabla\mathcal{J}(\boldsymbol{\theta}_k^{s+1};\boldsymbol{\theta}_k^{s+1})\|^2\right] + 2L^2\mathbb{E}\|\boldsymbol{\theta}_k^{s+1} - \widetilde{\boldsymbol{\theta}}^s\|^2$$

which means

$$\mathbb{E}[\|A\|] \leq \sqrt{\mathbb{E}[\|A\|^2]} \leq \sqrt{2\mathbb{E}\left[\|\nabla\mathcal{J}(\boldsymbol{\theta}_k^{s+1};\boldsymbol{\theta}_k^{s+1})\|^2\right] + 2L^2\mathbb{E}\|\boldsymbol{\theta}_k^{s+1} - \widetilde{\boldsymbol{\theta}}^s\|^2}$$

Applying Young's Inequality on rhs, for any positive $\lambda$ we have:

$$\mathbb{E}\|A\| \leq \frac{1}{\lambda}\left(\mathbb{E}\left[\|\nabla\mathcal{J}(\boldsymbol{\theta}_k^{s+1};\boldsymbol{\theta}_k^{s+1})\|^2\right] + L^2\mathbb{E}\left\|\boldsymbol{\theta}_k^{s+1} - \widetilde{\boldsymbol{\theta}}^s\right\|^2\right) + \frac{\lambda}{2} \tag{23}$$

Let $\lambda_0 = 4L_0\epsilon(L^2+1)$, we have:

$$\begin{aligned}
\mathbb{E}\|A\| &\leq \frac{1}{\lambda_0}\left(\mathbb{E}\left[\|\nabla\mathcal{J}(\boldsymbol{\theta}_k^{s+1};\boldsymbol{\theta}_k^{s+1})\|^2\right] + L^2\mathbb{E}\left\|\boldsymbol{\theta}_k^{s+1} - \widetilde{\boldsymbol{\theta}}^s\right\|^2\right) + \frac{\lambda_0}{2} \\
&\leq \frac{1}{4L_0\epsilon}\mathbb{E}\left[\|\nabla\mathcal{J}(\boldsymbol{\theta}_k^{s+1};\boldsymbol{\theta}_k^{s+1})\|^2\right] + \frac{1}{4L_0\epsilon}\mathbb{E}\left\|\boldsymbol{\theta}_k^{s+1} - \widetilde{\boldsymbol{\theta}}^s\right\|^2 + \frac{\lambda_0}{2}
\end{aligned} \tag{24}$$

According to the point-wise Lipschitzness in Lemma 5.2, we have:

$$\mathbb{E}[\|B\|] \leq \sqrt{\mathbb{E}[\|B\|^2]} \leq \sqrt{L_0^2\epsilon^2\mathbb{E}\left\|\boldsymbol{\theta}_k^{s+1} - \boldsymbol{\theta}^s\right\|^2} \leq \frac{\mathbb{E}\left\|\boldsymbol{\theta}_k^{s+1} - \widetilde{\boldsymbol{\theta}}^s\right\|^2}{4L_0\epsilon} + L_0^3\epsilon^3 \tag{25}$$

Thus, we have

$$\begin{aligned}
\mathbb{E}[\|v_k^{s+1}\|] &\leq \mathbb{E}\|A\| + \mathbb{E}\|B\| \\
&\leq \frac{\mathbb{E}\|\nabla\mathcal{J}(\boldsymbol{\theta}_k^{s+1};\boldsymbol{\theta}_k^{s+1})\|^2}{4L_0\epsilon} + \left(\frac{1}{4L_0\epsilon} + \frac{1}{4L_0\epsilon}\right)\mathbb{E}\left\|\boldsymbol{\theta}_k^{s+1} - \widetilde{\boldsymbol{\theta}}^s\right\|^2 + \frac{\lambda_0}{2} + L_0^3\epsilon^3 \\
&= \frac{\mathbb{E}\|\nabla\mathcal{J}(\boldsymbol{\theta}_k^{s+1};\boldsymbol{\theta}_k^{s+1})\|^2}{4L_0\epsilon} + \frac{1}{2L_0\epsilon}\mathbb{E}\left\|\boldsymbol{\theta}_k^{s+1} - \widetilde{\boldsymbol{\theta}}^s\right\|^2 + \left(\frac{\lambda_0}{2} + L_0^3\epsilon^3\right)
\end{aligned} \tag{26}$$

Finally, we plug Eqn. 26 into Eqn. 21, and then plug Eqn. 21 into Eqn. 20. This leads to the inequality relationship between $\mathcal{J}(\boldsymbol{\theta}_{k+1}^{s+1}; \boldsymbol{\theta}_{k+1}^{s+1})$ and $\mathcal{J}(\boldsymbol{\theta}_k^{s+1}; \boldsymbol{\theta}_k^{s+1})$.

$$
\mathbb{E}\left[\mathcal{J}(\boldsymbol{\theta}_{k+1}^{s+1}; \boldsymbol{\theta}_{k+1}^{s+1}) + c_{k+1}\|\boldsymbol{\theta}_{k+1}^{s+1} - \widetilde{\boldsymbol{\theta}}^s\|^2\right] \leq \mathbb{E}\left[\mathcal{J}(\boldsymbol{\theta}_k^{s+1}; \boldsymbol{\theta}_k^{s+1}) + c_k\|\boldsymbol{\theta}_k^{s+1} - \widetilde{\boldsymbol{\theta}}^s\|^2\right]
$$
$$
- (\Gamma_k - \frac{\gamma_k}{4}) \cdot \mathbb{E}[\|\nabla\mathcal{J}(\boldsymbol{\theta}_k^{s+1}; \boldsymbol{\theta}_k^{s+1})\|^2]
$$
$$
+ (L_0^4\epsilon^4 + 2L_0^2\epsilon^2 L^2 + 2L_0^2\epsilon^2)\gamma_k
$$

which is exactly what we want. □

## D  PROOF OF THM. 5.4

*Theorem.* Let the total rounds be $T$ and the epochs be $S = \lceil T/m \rceil$. For the last epoch, $c_m = 0$. Define $\Delta_0$ and $\Delta_1$ as follows:

$$
\Delta_0 \triangleq \mathcal{J}(\boldsymbol{\theta}_0^0; \boldsymbol{\theta}_0^0) - \ell_*; \quad \Delta_1 \triangleq \frac{(L_0^4\epsilon^4 + 2L_0^2\epsilon^2 L^2 + 2L_0^2\epsilon^2)\sum_{s=0}^{S}\sum_{k=0}^{m}\gamma_k}{T\Gamma}
$$

where $\Gamma > 0$ is guaranteed to exist as a constant and is the lower bound of $\Gamma_k - \frac{\gamma_k}{4}$. Then we have:

$$
\frac{1}{T}\sum_{s=0}^{S-1}\sum_{k=0}^{m-1}\mathbb{E}\|\nabla\mathcal{J}(\boldsymbol{\theta}_k^{s+1}; \boldsymbol{\theta}_k^{s+1})\|^2 \leq \frac{\Delta_0}{T\Gamma} + \Delta_1 \tag{27}
$$

Thus, Alg. 1 can achieve $\mathcal{O}(\frac{1}{T})$ convergence rate and the error neighborhood $\Delta_1$ is $\mathcal{O}(\epsilon^2 + \epsilon^4)$.

*Proof.* Note the $\gamma_t$ is just an abbreviation of the learning rate at global round $s$ and local round $k$. Using Lemma 5.3 (Eqn. 6) and telescoping:

$$
\sum_{s=0}^{S-1}\sum_{k=0}^{m-1}(\Gamma_k - \frac{\gamma_k}{4}) \cdot \mathbb{E}[\|\nabla\mathcal{J}(\boldsymbol{\theta}_k^{s+1}; \boldsymbol{\theta}_k^{s+1})\|^2] \leq (\widetilde{R}_0^1 - \widetilde{R}_m^S) + (L_0^4\epsilon^4 + 2L_0^2\epsilon^2 L^2 + 2L_0^2\epsilon^2)\sum_{t=0}^{T-1}\gamma_t \tag{28}
$$

Thus,

$$
\frac{1}{T} \cdot \sum_{s=0}^{S-1}\sum_{k=0}^{m-1}\mathbb{E}[\|\nabla\mathcal{J}(\boldsymbol{\theta}_k^{s+1}; \boldsymbol{\theta}_k^{s+1})\|^2] \leq \frac{\Delta_0}{T\Gamma} + \Delta_1
$$

There is one remaining step: prove there exists values of $\gamma_k, \beta_k$ to let $\Gamma > 0$. We first begin with a lemma:

**Lemma D.1.** *For any $\eta > 0$ and $\beta_k$ is some constant $\beta$, there exists a sequence of learning rate $\{\gamma_k\}_{k=1}^m$ to let $c_k \leq \eta L_0 \epsilon$ hold for any $k$.*

*Proof.* We can start from $c_m = 0$ and are given the relationship between $c_k$ and $c_{k+1}$:

$$
c_k - \frac{1}{2}\gamma_k = c_{k+1}(1 + \gamma_k\beta_k + 2L_0\epsilon\gamma_k) + (2L^2 + L_0^2\epsilon^2)(L\gamma_k^2 + 2c_{k+1}\gamma_k^2) + \frac{\beta_k}{2}L_0\epsilon\gamma_k
$$

For convenience, define $A_k = 1 + \gamma_k\beta + 2L_0\epsilon\gamma_k + (4L^2 + 2L_0^2\epsilon^2)\gamma_k^2$, $B_k = (2L^2 + L_0^2\epsilon^2)L\gamma_k^2 + \frac{\beta}{2}L_0\epsilon\gamma_k$. Then, the recurrence can be rewritten as $c_k = \frac{1}{2}\gamma_k + A_k c_{k+1} + B_k$. Next, we use induction to prove that for any $\eta > 0$, there exists a sequence of $\{\gamma_k\}_{k=1}^m$ (depending only on $L$, $L_0$, $\epsilon$, and $\beta$) such that for all indices $k$ we have $c_k \leq \eta L_0 \epsilon$.

Since we start from $c_m = 0$, we have $c_m = 0 \leq \eta\, L_0\epsilon$ holding for any positive $\eta$. Next, assume for some $k+1$, $c_{k+1} \leq \eta\, L_0\epsilon$. Then we can work out $c_k$ as:

$$c_k = \frac{1}{2}\gamma_k + A_k\, c_{k+1} + B_k \leq \frac{1}{2}\gamma_k + A_k\Big(\eta\, L_0\epsilon + C\,\gamma_k\Big) + B_k.$$

Since we can adjust $\gamma_k$ to be small, we can expand $A_k = 1 + \gamma_k\beta + 2L_0\epsilon\,\gamma_k + \mathcal{O}(\gamma_k^2)$ and $B_k = \frac{\beta}{2}L_0\epsilon\,\gamma_k + \mathcal{O}(\gamma_k^2)$. Thus,

$$c_k \leq \frac{1}{2}\gamma_k + \Big(1 + \gamma_k\beta + 2L_0\epsilon\,\gamma_k\Big)\eta\, L_0\epsilon + \frac{\beta}{2}L_0\epsilon\,\gamma_k + C_1\gamma_k^2$$

$$= \eta\, L_0\epsilon + \left\{\frac{1}{2} + \eta\, L_0\epsilon(\beta + 2L_0\epsilon) + \frac{\beta}{2}L_0\epsilon\right\}\gamma_k + C_1\gamma_k^2$$

$$= \eta\, L_0\epsilon + C_0\gamma_k + C_1\gamma_k^2.$$

where $C_0, C_1$ are constants represented by $L, L_0, \epsilon, \beta$. This means for arbitrarily small number $\omega_k$, we can simply set $\gamma_k$ to be $\min\{1, \frac{\omega_k}{C_0+C_1}\}$ to let $c_k \leq (\eta + \omega_k)L_0\epsilon$. Then from $c_m \leq \eta L_0\epsilon$ we can derive that $c_k \leq (\eta + \Sigma_{i=0}^{m}\omega_i)L_0\epsilon$ holds. Finally, for any $\eta$, we can surely find a $\widetilde{\eta} > 0$ to let $\widetilde{\eta} + \Sigma_{i=0}^{m}\omega_i < \eta$, so we prove that $c_k \leq \eta L_0\epsilon$. $\qquad\square$

Next, we prove that with the learning rates in Lemma D.1, then $\Gamma > 0$. We first assume $\gamma_k < \frac{1}{8L + \frac{8}{3}L_0\epsilon}$.

$$\Gamma_k = \gamma_k - \frac{c_{k+1}\gamma_k + \frac{1}{2}L_0\epsilon\gamma_k}{\beta} - 2L\gamma^2 - 4c_{k+1}\gamma_k^2.$$

Using the bound $c_{k+1} \leq \eta L_0\epsilon$ and setting $\beta = 4L + \frac{4}{3}L_0\epsilon$, we obtain

$$\Gamma_k \geq \gamma_k - \frac{(\eta L_0\epsilon + \frac{1}{2}L_0\epsilon)\gamma_k}{4L + \frac{4}{3}L_0\epsilon} - 2L\gamma_k^2 - 4\eta L_0\epsilon\gamma_k^2$$

$$= \gamma_k - \frac{(\eta + \frac{1}{2})L_0\epsilon\gamma_k}{4L + \frac{4}{3}L_0\epsilon} - 2L\gamma_k^2 - 4\eta L_0\epsilon\gamma_k^2.$$

Then we can bound the quadratic terms as follows:

- $2L\gamma_k^2 \leq 2L\left(\frac{1}{8L + \frac{8}{3}L_0\epsilon}\right)\gamma_k < 2L \cdot \frac{1}{8L}\gamma_k = \frac{\gamma_k}{4}$

- $4\eta L_0\epsilon\gamma_k^2 \leq \frac{2\eta L_0\epsilon\gamma_k}{4L + \frac{4}{3}L_0\epsilon}$ (since $\gamma_k \leq \frac{1}{8L + \frac{8}{3}L_0\epsilon}$).

Thus,

$$\Gamma_k \geq \gamma_k - \frac{(\eta + \frac{1}{2})L_0\epsilon\gamma_k}{4L + \frac{4}{3}L_0\epsilon} - \frac{\gamma_k}{4} - \frac{2\eta L_0\epsilon\gamma_k}{4L + \frac{4}{3}L_0\epsilon}.$$

Combine the two fractions involving $L_0\epsilon$:

$$\Gamma_k \geq \gamma_k\left[1 - \frac{1}{4} - \frac{(3\eta + \frac{1}{2})L_0\epsilon}{4L + \frac{4}{3}L_0\epsilon}\right].$$

To bound $\Gamma_k \geq \frac{\gamma_k}{4}$, we need:

$$1 - \frac{1}{4} - \frac{(3\eta + \frac{1}{2})L_0\epsilon}{4L + \frac{4}{3}L_0\epsilon} \geq \frac{1}{4}.$$

This means if $\eta \leq \frac{4L + \frac{1}{3}L_0\epsilon}{6L_0\epsilon}$, we have $\Gamma \geq \frac{\gamma_k}{4}$. Finally, denote $\frac{4L + \frac{1}{3}L_0\epsilon}{6L_0\epsilon}$ as $\eta_0$, then consider $\eta = \eta_0$ in Lemma D.1, we can let $\omega_k = \omega = \frac{\eta_0}{2m}$ to ensure $\Sigma_{i=1}^m \omega_k < \eta_0$. Then we can set $\gamma = \min\{1, \frac{1}{8L + \frac{8}{3}L_0\epsilon} \frac{\omega}{C_0 + C_1}\}$ and set $\gamma_k = \gamma$ to let so that Lemma D.1 also holds. Aggregating all these together, we finally prove that $\Gamma \geq \min \frac{\gamma_k}{4} = \frac{\gamma}{4}$ which is a constant, resulting in the $\mathcal{O}(\frac{1}{T})$ convergence given that $\gamma$ is a constant.

Finally, consider $\Delta_1$ when $\Gamma \geq \frac{\gamma}{4}$, it is obvious it reduces to $L_0^2\epsilon^2 + L_0^4\epsilon^4$ multiplied by some constant. When $\epsilon$ is smaller than 1, the error neighborhood is $\mathcal{O}(\epsilon^2 + \epsilon^4)$. □

## E    PROOF OF COROLLARY 5.5

*Corollary.* Assume we have a finite distribution setting where the population consists of $n$ samples in total and $\alpha \in (0, 1)$. Then when we have fixed parameters at each round, i.e., $\gamma_k = \gamma = \mathcal{O}(\frac{1}{n^\alpha}), \beta_k = \beta = \mathcal{O}(n^{\frac{\alpha}{2}}), m = \mathcal{O}(n^{\frac{\alpha}{2}})$, then we have $\Gamma = \mathcal{O}(\frac{1}{n^\alpha})$, and the IFO complexity is $\mathcal{O}(\frac{(n^\alpha + n^{1 + \frac{\alpha}{2}})\Delta_0}{\delta})$ to achieve $\mathcal{O}(\delta)$-SPS in addition to the error neighborhood.

*Proof.* From the Lemma D.1 and the proof in App. D, we already know that when $m = \mathcal{O}(n^{\frac{\alpha}{2}})$, any $\gamma_k \leq \gamma = \mathcal{O}(\frac{\eta_0}{2m(C_0 + C_1)})$ will make $\Gamma > \frac{\gamma_k}{2}$. Since $C_0, C_1$ are all $\mathcal{O}(\beta) = \mathcal{O}(n^{\frac{\alpha}{2}})$, then we know $\gamma = \mathcal{O}(\frac{1}{n^\alpha})$ is a plausible choice to satisfy Lemma D.1 to let $\Gamma > \frac{\gamma}{2}$. Moreover, $\Gamma > \frac{\gamma}{2} = \mathcal{O}(\frac{1}{n^\alpha})$ With this fact, we let $\frac{\Delta_0}{T\Gamma} \leq \delta$ to get $T$ should be $\mathcal{O}(\frac{n^\alpha \Delta_0}{\delta})$. Since for each $m$ rounds (1 epoch), we would have $2m + n$ evaluations of gradients. So the average sample complexity at each round is $2 + \frac{n}{m}$. Then, the final IFO complexity is:

$$T(2 + \frac{n}{m}) = \mathcal{O}\left(\frac{\Delta_0 n^\alpha (2 + n^{1 - \frac{\alpha}{2}})}{\delta}\right) = \mathcal{O}(\frac{(n^\alpha + n^{1 + \frac{\alpha}{2}})\Delta_0}{\delta})$$

□

## F    ADDITIONAL EXPERIMENTAL RESULTS

**Additional experimental settings.** We conduct all the experiments on a server which has two Intel Xeon 6326 CPUs and a Nvidia A6000 GPU. We implement our code using the pytorch of version 2.4.1 and seed 2024. In the PP settings, each training requires extensive steps to model distributional shifts, which significantly increases the computational cost, especially on large-scale datasets like CIFAR-10. We are only able to run one experiment on CIFAR-10, but we include experiments using 3 random seeds for the MNIST dataset as follows.

**Additional experiments on CIFAR-10 and Credit dataset.** We additionally report error bars for Credit and CIFAR-10 (with 10% of the data randomly selected) by rerunning each experiment with three different random seeds. The results are shown in Fig. 3 and Fig. 4. In Fig. 4, the error bars are relatively large compared to the gap between SPRINT and SGD-GD, which is expected since the Credit dataset is relatively small and SGD-GD can already converge fairly quickly. Nevertheless, SPRINT consistently maintains an advantage over SGD-GD across seeds.

**Experiments on MNIST dataset.** We use the MNIST dataset (Deng, 2012) and train a two-layer MLP to predict the 10 possible digits. We randomly sample approximately 12000 images and train the model using a learning rate set at 0.003. Similar to retention dynamics (Hashimoto et al., 2018; Jin et al., 2024; Zhang et al., 2019), we assume the class distribution will change based on the performance of the current model. The fraction in class $c$ at iteration $k$ of $p_c^{k+1} = \frac{e^{-\alpha \ell_c^k}}{\sum_{c'} e^{-\alpha \ell_{c'}^k}}$, where $\ell_c^k$ is the loss expectation of class $c$ in iteration $k$. This means the image class with larger

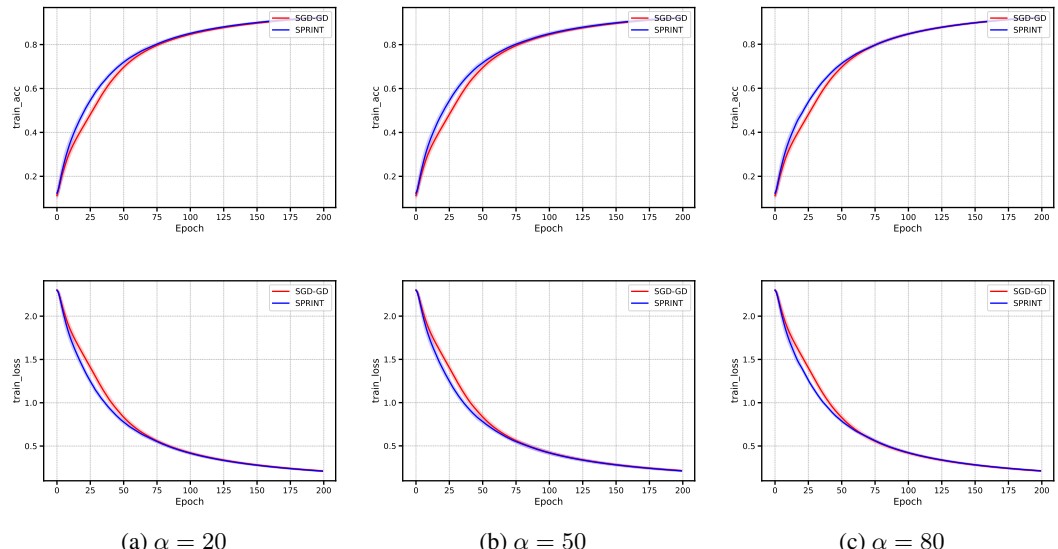

Figure 3: **CIFAR-10** dataset: training accuracy (first row) and training loss (second row) of SGD-GD and SPRINT under different $\alpha$. Larger $\alpha$ means more intense retention dynamics and the shadow demonstrates standard errors.

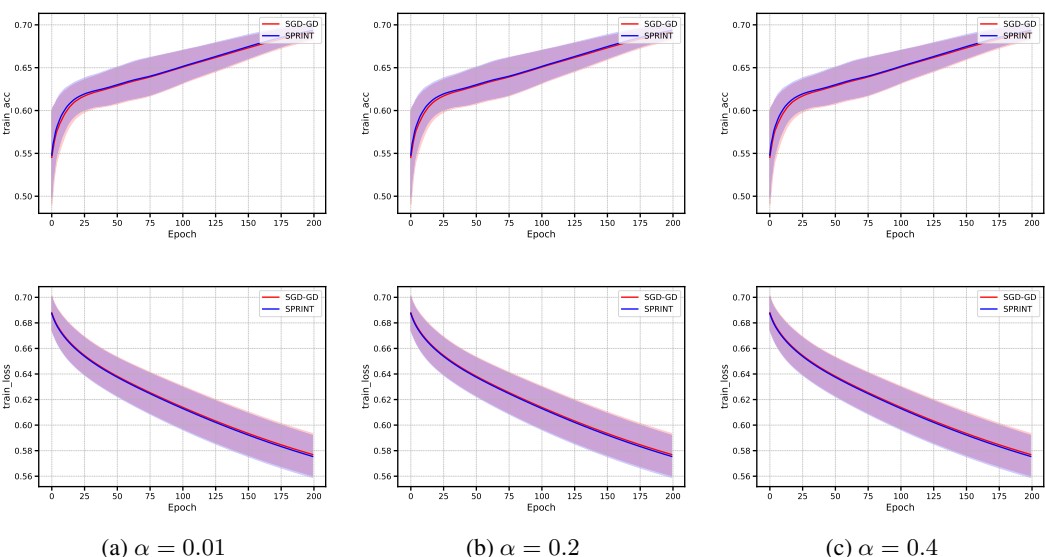

Figure 4: **Credit** dataset: training accuracy (first row) and training loss (second row) of SGD-GD and SPRINT under different $\alpha$. Larger $\alpha$ means more intense strategic behaviors and the shadow demonstrates standard errors.

loss now will less likely to appear at the next iteration (Zhang et al., 2019). We train the model for 80 epochs with cross entropy loss and visualize the accuracy and training loss curve in Fig. 5. We use three random seeds, which are 2024, 2025, 2026, to run the experiments. The $\alpha$ is set at $20, 50, 80$ to simulate different magnitudes of performative effects. In Fig. 5, the training loss of SPRINT consistently converges faster, which is also reflected in the training accuracy.

Moreover, we also iterate through different learning rates and verify that SPRINT consistently outperforms SGD-GD. In Fig. 7, we plot the dynamics of training loss and accuracy when learning rate in $\{0.001, 0.003, 0.1\}$ and the results demonstrate the competence of SPRINT .

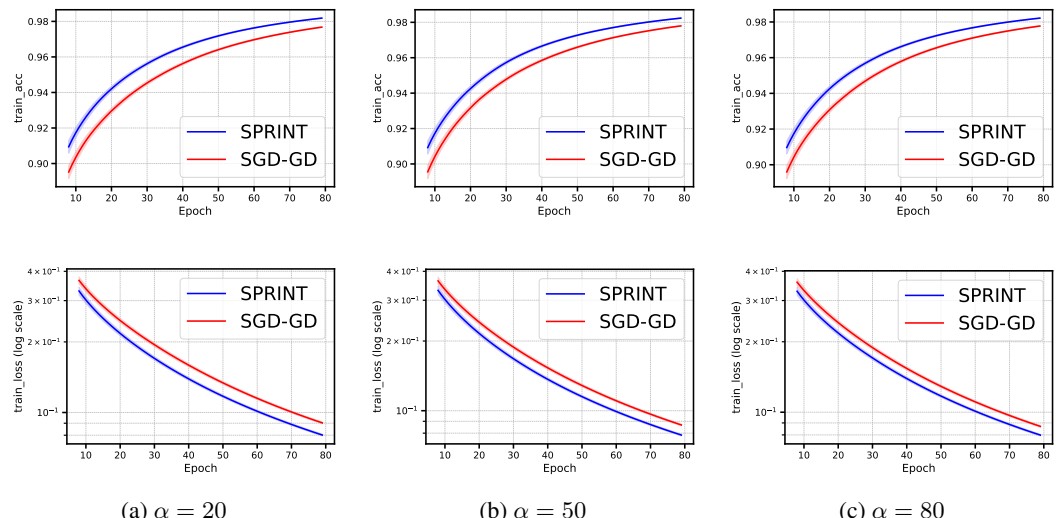

Figure 5: **MNIST** dataset: training accuracy (first row) and training loss (second row) of SGD-GD and SPRINT under different $\alpha$. Larger $\alpha$ means more intense retention dynamics and the shadow demonstrates standard errors.

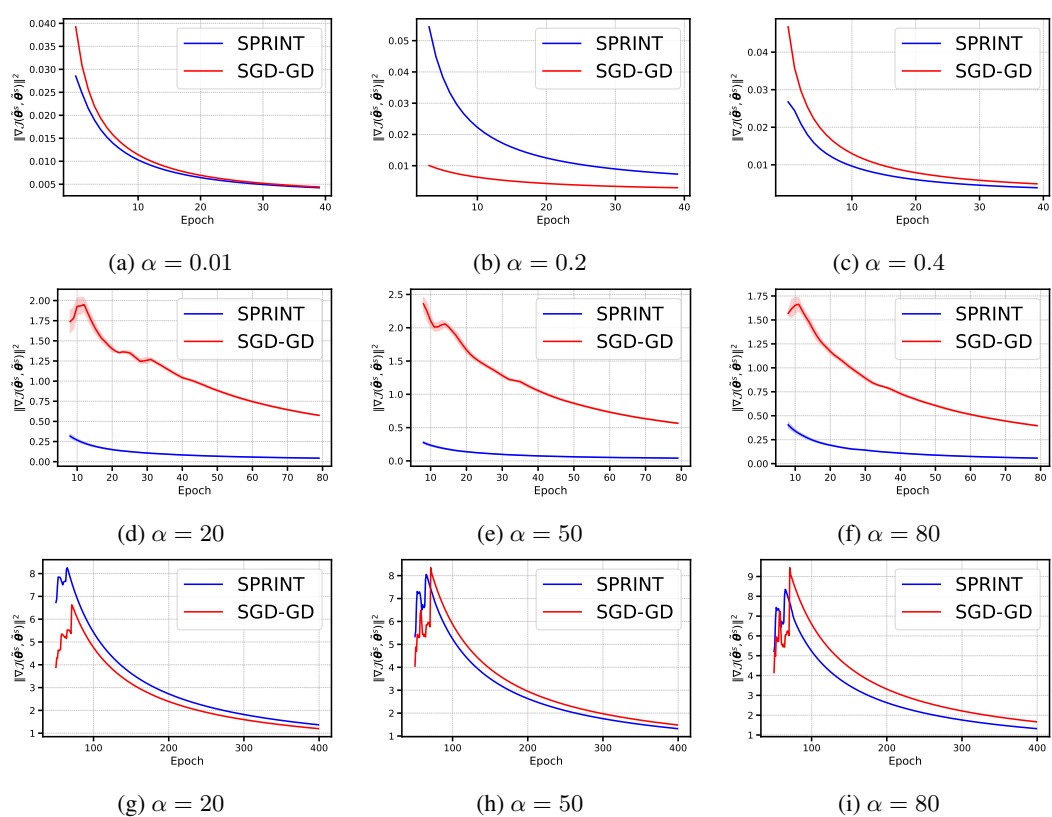

Figure 6: Squared gradient norm $\|\nabla \mathcal{J}(\widetilde{\boldsymbol{\theta}}^s; \widetilde{\boldsymbol{\theta}}^s)\|^2$. From Up to down are **Credit** dataset, **MNIST** dataset, and **CIFAR-10** dataset.

**The average of squared gradient norm.** We visualize the cumulative average of the squared gradient norm at the end of each epoch $\|\nabla \mathcal{J}(\widetilde{\boldsymbol{\theta}}^s; \widetilde{\boldsymbol{\theta}}^s)\|^2$ of SGD-GD and SPRINT in all 3 settings as specified in Sec. 6 shown in Fig. 6. Notably, in MNIST setting (Fig. 6d to Fig. 6f), SPRINT con-

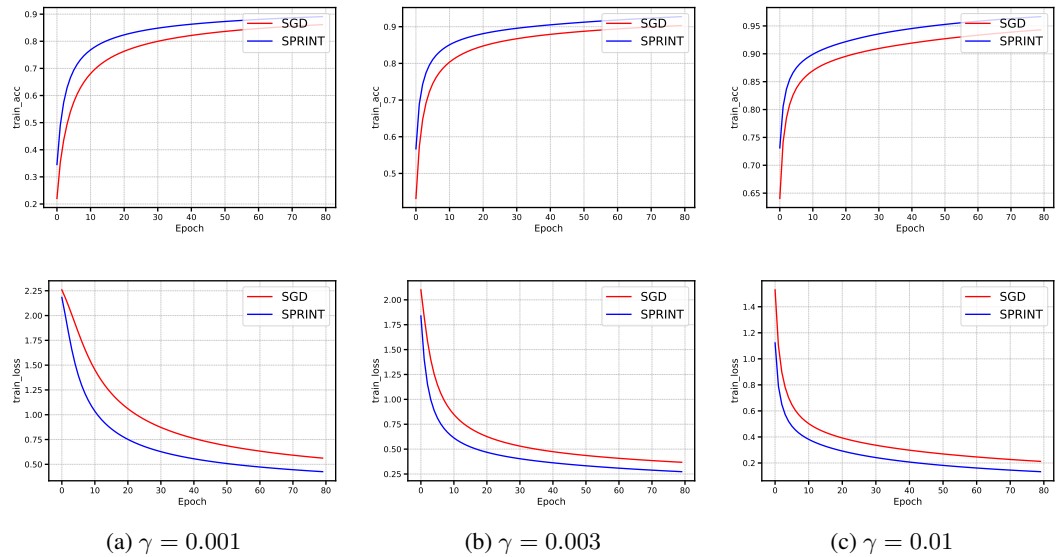

(a) $\gamma = 0.001$        (b) $\gamma = 0.003$        (c) $\gamma = 0.01$

Figure 7: **MNIST** dataset: training accuracy (first row) and training loss (second row) of SGD-GD and SPRINT under different learning rate $\gamma$.

stantly has much smaller squared gradient norm, demonstrating the effectiveness of our algorithm. In Credit setting (Fig. 6a to Fig. 6c), SPRINT constantly has a smaller squared gradient norm when $\alpha = 0.01$ and $0.4$. When $\alpha = 0.2$, SPRINT has a much worse initial point than SGD-GD, resulting in the large squared gradient norm at first but similar norm at the end. Similarly, in CIFAR-10 setting (Fig. 6g to Fig. 6i), SPRINT has a smaller squared gradient norm when $\alpha = 50$ and $80$. When $\alpha = 20$, SPRINT has a much worse initial point than SGD-GD, resulting in the large squared gradient norm at first but similar norm at the end.

Note that Thm. 5.4 is on the expectation of the squared gradient norm, but we are only able to conduct one experiment for each setting due to the limitation of computational resource. Thus, we can have some bad initialization point of SPRINT to influence the results.

## G    EXTEND SPRINT TO INFINITE SUM SETTINGS

Our current finite-sum analysis avoids bounded variance by using a full gradient. For the population expectation minimization setting, we may have two plausible ways based on our current proof framework:

1. We may use a minibatch of samples as the gradient snapshot, but we do need the bounded variance assumption to obtain SVRG-type rates.

2. We can consider methods similar to Jothimurugesan et al. (2018), where the minibatch is growing and a control variate method is used to keep the gradient variance bounded. In this way, we do not need an explicit bounded variance assumption.

With the above methods and replacing the full gradients with mini-batch gradients, all other derivations should be similar to our current work. However, some nuanced constant parameters tuning may be needed to take care of the additional error produced by the mini-batch gradients.

## H    COMPARISON OF IFO COMPLEXITY BETWEEN SPRINT AND SGD-GD

According to Theorem 1 of Li & Wai (2024), we can directly get the IFO Complexity of SGD-GD. When $n$ is not too large and when we require $\delta$ to be small, SPRINT will have much smaller complexity. Meanwhile, when $\sigma_0$ is large or even intractable (e.g., heavy-tailed/fat-tailed noise

settings (Gurbuzbalaban et al., 2021)), SGD-GD can also have extremely large or even infinite IFO Complexity, while the complexity of SPRINT is independent of the variance parameter $\sigma_0$.

In Corollary 5.5 and Appendix H, we have derived the IFO complexity of SPRINT to be $\mathcal{O}(\frac{(n^{\alpha}+n^{1+\frac{\alpha}{2}})\Delta_0}{\delta})$. Compared to the complexity of SGD-GD is $\mathcal{O}(\frac{1}{\delta^2})$, we know there is a trade-off between population size $n$ and the error bound in additional to the error neighborhood $\delta$. When the population size is very large while the error $\delta$ is not too small, the computation of SPRINT can be slower than SGD-GD.

