# OpenReview forum: "SPRINT: Stochastic Performative Prediction With Variance Reduction"
_ICLR.cc/2026/Conference — Submitted to ICLR 2026_

### Official Review · Reviewer_fJGn · 2025-10-25

**Soundness:** 2
**Presentation:** 2
**Contribution:** 2
**Rating:** 2
**Confidence:** 4

**Summary:**

This paper proposes SPRINT for performative prediction: an SVRG-style, snapshot-based variance-reduction method that tames noise from deployment-induced distribution shifts, proves fast and stable convergence without bounded-variance assumptions, and empirically outperforms SGD-GD on Credit, MNIST, and CIFAR-10—especially when feedback effects are strong.

**Strengths:**

1. The method converges faster and more stably than SGD-GD, reaching a final neighborhood that does not depend on gradient variance.

2. It operates under weaker assumptions by avoiding the bounded-variance requirement and explicitly addressing deployment-induced distribution shift.

3. It delivers stronger empirical performance, achieving lower loss and steadier training on Credit, MNIST, and CIFAR-10, with especially large gains when performative effects are strong.

**Weaknesses:**

Equation numbering is inconsistent (two equations labeled as Eq. 14).

The roles of $c_k$, $\beta_k$, and $\gamma_k$ are not clearly explained. Their definitions and interactions in the update rule should be clarified.

It is unclear whether Assumption 3.3 could be relaxed; this assumption seems restrictive for broader stochastic settings.

The paper should also discuss scenarios where SPRINT might underperform relative to SGD-GD (e.g., small population, low-variance cases) and analyze why.

Although hyperparameters are reported, the lack of details on the hyperparameter search process raises fairness concerns for baseline comparison.

The claimed “variance reduction” effect is not empirically validated—there is no report of gradient variance, convergence variance, or ablation removing the VR component.

 The experiments only report training loss; test accuracy should also be presented, especially given SPRINT’s claimed robustness under data distribution shifts.

Quantitative data on computational and memory costs are missing.

Experiments rely on small MLP and CNN architectures; results on more realistic models such as ResNet or VGG would strengthen the paper.

 The sensitivity of SPRINT to the learning rate remains unclear—this should be evaluated experimentally.

 It is also important to clarify whether SPRINT introduces additional computational or memory overhead compared to SGD-GC.

**Questions:**

see  Weaknesses

---

> ### Author Response · Authors · 2025-11-19
> **Response to Reviewer fJGn**
>
> Thanks for your comments. We address your questions and concerns as follows:
>
> > Equation numbering is inconsistent
>
> We fix it in the modified draft.
>
> > $c_k, \beta_k, \gamma_k$
>
> - $\gamma_k$ is the learning rate used in Algorithm 1;
> - $\beta_k$ and $c_k$ are *analytical constants* introduced only in the convergence proof (see Lemma 5.1–5.3). They are chosen to satisfy specific recursive relations which guarantee the descent property of the Lyapunov function and ensure that $\Gamma_k > 0$, leading to the $O(\frac{1}{T})$ convergence rate in Theorem 5.4. These parameters therefore play a purely *theoretical* role and are not part of the algorithmic implementation.
>
> > Assumption 3.3
>
> The Lipschitz condition is a **standard and mild regularity assumption** which has also been used in SGD in PP settings [Li & Wai, 2024]. It serves a purely technical role to bound the difference between consecutive performative risks in Lemma 5.2 and establishing convergence guarantees. In practice, this assumption is easily satisfied: common smooth losses such as squared, logistic, or cross-entropy loss are Lipschitz in their data argument on any **bounded domain**, and empirical datasets are always normalized or constrained to such domains. Therefore, the condition does not impose any additional burden beyond what is already standard in stochastic optimization. Even if the loss is only *locally* Lipschitz on bounded domain, the same analysis holds up to constant factors. Hence, Assumption 3.3 is not restrictive.
>
> > The paper should also discuss scenarios where SPRINT might underperform relative to SGD-GD (e.g., small population, low-variance cases) and analyze why.
>
> - In Corollary 5.5 and Appendix H, we have derived the IFO complexity of SPRINT to be $\mathcal{O}(\frac{(n^{\alpha} + n^{1+\frac{\alpha}{2}})\Delta_0}{\delta})$. By contrast, the complexity of SGD-GD is $\mathcal{O}(\frac{1}{\delta^2})$, which means there is a trade-off between population size $n$ and the error bound in additional to the error neighborhood $\delta$. When the population size is very large while the error $\delta$ is not too small, the computation of SPRINT can be slower than SGD-GD. **We have added this sentence to Appendix H in modified draft**.
>
> - Note that this trade-off between sample complexity and approximation error also arises when comparing SGD with variance reduction methods in standard stochastic optimization, even without performative effects. Nevertheless, variance reduction remains beneficial, as it avoids frequent full population access while eliminating the need for bounded variance assumptions.

---

> ### Author Response · Authors · 2025-11-19
>
> > Experiments
>
> **First, we hope to reiterate our main contribution is theoretical**. Also, we believe our experiments are more comprehensive than previous literature on PP (e.g., Perdomo et al., 2021, Li & Wai, 2024) which mostly used synthetic dataset, credit/spam dataset without touching on larger CV dataset such as MNIST/CIFAR. Meanwhile, to the best of our knowledge, all important previous literature on PP and SVRG (e.g., Perdomo et al., 2021, Izzo et al., 2021, Li & Wai, 2024) did not perform experiments on complex NN structures such as Resnet / VGG.
>
> **To provide more convincing evidence, we have added the following experiments in modified draft:**
>
> - hyperparameter search process
>
> We add an experiment with different learning rates and compare the loss/accuracy dynamics of SPRINT and SGD-GD in **Figure 7, Appendix F** of the modified draft.
>
> - gradient variance, convergence variance
>
> We have provided the error bars of gradient norms for MNIST data in Figure 6, Appendix F. This is because in PP settings, each training requires extensive steps to model distributional shifts, which significantly increases the computational cost, especially on large-scale datasets like CIFAR-10. We are only able to run one experiment on CIFAR-10, but we include experiments using $3$ random seeds for MNIST dataset.
>
> To provide more convincing evidence, we added error bars using different random seeds for Credit and CIFAR-10 (10% data randomly selected) by rerunning the experiments on 3 different seeds. Results are in **Figure 3 and 4, Appendix F** of the modified draft.
>
>
> - Test accurcy
>
> The concept of “test accuracy” does not directly apply to the performative prediction framework. Our work studies the optimization dynamics and theoretical convergence of models whose deployment continuously reshapes the data distribution $D(\theta)$. In this setting, there is no fixed test distribution, where evaluation naturally concerns the post-deployment accuracy which we already report. Previous literature [Li & Wai, 2024, Perdomo et al., 2021] also do not incorporate test accuracy.
>
>
> - Quantitative data on computational and memory costs are missing. It is also important to clarify whether SPRINT introduces additional computational or memory overhead compared to SGD-GC.
>
> We already provided IFO complexity in Corollary 5.5 and compare SPRINT to SGD-GD in **Appendix H**. **We also added more details there in the modified draft.**
>
> - The sensitivity of SPRINT to the learning rate remains unclear—this should be evaluated experimentally.
>
> We provide an additional experiment in MNIST data with different learning rates and compare the loss/accuracy dynamics of SPRINT and SGD-GD in **Figure 7, Appendix F** of the modified draft. The results demonstrate SPRINT outperfoms SGD-GD under different learning rates.

---

### Official Review · Reviewer_Dyhx · 2025-10-27

**Soundness:** 2
**Presentation:** 2
**Contribution:** 2
**Rating:** 4
**Confidence:** 4

**Summary:**

The paper proposes using the SVRG method for performative prediction, that is, minimizing a loss where the data distribution depends on the parameters being optimized. By adapting the SVRG proof to this setting, the author can establish a 1/T rate to a stationary point. Their proof also drops a previously used "bounded variance assumption".

**Strengths:**

One clear strength I see is that, as compared to Li & Tai 2024, these new convergence results do not rely on an explicit bounded variance assumption. Though implicitly the variance is bounded by the remaining assumptions. In any case, they use fewer assumptions.

**Weaknesses:**

The main weakness I see, and the one that my current stance hinges on, is the contradiction between the finite sum structure, and having a parameter dependent distribution.

**Finite sum breaks the data dependent assumption**.  How can you assume finite sum and that $z_i$ depends on $\theta$? By assuming the full loss has a finite sum structure, you loose the functional dependency on $\theta$, and you can no longer differentiate through a discrete sample. That is, having
$$ \nabla \mathcal{J}(\theta;\theta)  = \frac{1}{n} \sum_{I=1}^n \nabla \ell (\theta; z_i)$$ you essentially loose the dependence between the samples $z_i$'s and $\theta$. If not, how exactly are you now parametrizing the sampling of $z_t \in \{z_1, \ldots, z_n\}$ from this finite set so that it depends on $\theta_t$? This is why the related fields I mention below all have to contend with true expectations (infinite population). I read your Appendix G that briefly mentions how to avoid this issue, but it is not enough. You must contend with infinite samples, otherwise there is a contradiction between finite sum and samples depending on $\theta$.



**Limited related work.** This problem as stated

$$ \min\_{\theta \sim \mathcal{D}\_{\theta}} \mathbb{E}\_{z \sim \mathcal{D}\_{\theta}}[\ell(\theta, z)] $$

Is verbatim the same problem stated in numerous fields, including RL, Optimal control, Variational Inference, and more. It has been an ongoing challenge in all of these fields to establish a non-asymptotic converges of SGD and variance reduced versions. These different fields have to deal with the exact same issues you address, such as having biased estimates of the gradient, assuming smoothness over a support of $z$, and analogous variance estimates. Because of this, you should extend your background section beyond predictive performance, and explain to the reader why this hasn't been done in RL/Control/VI before. A few standard references in convergence of SGD in RL below, but you should cast a wider net:

Lin Xiao, On the Convergence Rates of Policy Gradient Methods,  Journal of Machine Learning Research 23 (2022) 1-36

Alekh Agarwal, Sham M. Kakade, Jason D. Lee, and Gaurav Mahajan. On the theory of policy gradient methods: optimality, approximation, and distribution shift. J. Mach. Learn. Res. 22, 1, Article 98 (January 2021), 76 pages.

Rui Yuan, Robert M. Gower Alessandro Lazaric, A general sample complexity analysis of vanilla policy gradient, AISTATS 2022




**Variance independent bounds** This emphasis on not having "variance" appear in your bound is perhaps overplayed. This is because you assume the loss is $L_0$-Lipschitz, which bounds the gradient norm, and thus plays now the role of "variance" in your bounds. To be specific, using Lemma 5.2 from Li & Wai, assuming $L_0$-Lipschitz loss and $\epsilon$--sensitivity, you have immediately that Assumption 3.4 holds with $\sigma_0^2 = 2L_0^2 + 2\epsilon^2 L_0^2$ and $\sigma_1^2 =0.$ As such, I think this emphasis on not having dependency on variance is a bit confusing. A stronger message is that you simply rely on less assumptions (you do not explicitly assume  Assumption 3.4.
*Questions*

**Questions:**

Your experiments report training accuracy. Is there a corresponding notion of test accuracy here for  performative prediction? I'm guessing there is no such notion. Which make me think that online learning is a better paradigm for this problem.



**Minor issues**

1. Assumption 1: You should state $\forall z \in \mathcal{Z}.$

2. Definition 1. Because you evaluate the full batch gradient on the same $\theta$ as the one from which $z$ is sampled, this condition is exactly the standard $\delta$ stationarity condition, why has it been renamed to "$\delta$-Stationary Performative Stable Solution"? I find this confusing.


3. Formal statement of assumptions. You need to be crystal clear about your assumptions. In a formal Lemma, to be clear, you should always include the assumptions in the statement. For instance, does each Lemma 5.1 and 5.3 use all of your assumptions or only a subset? These things have to be clear for a theory paper.

4.  On a subjective note here, the name "Performative Prediction" is a very unfortunate name. It does not give any hint that this is an optimization problem where the data distribution is parameter dependent. This naming will make it harder for researchers in RL/control/VI working on this exact problem to find these works.

---

> ### Author Response · Authors · 2025-11-19
> **Response to Reviewer Dyhx**
>
> Thanks for the insightful comments. We address each weakness and question as follows:
>
>
> > **W1:** Finite sum breaks the data dependence assumption: assuming a finite-sum structure contradicts the parameter-dependent distribution $D(\theta)$, since discrete samples cannot depend on $\theta$.
>
> **Answer:**
> We kindly note that a discrete distribution can still be model-dependent, as shown below:
>
> $$D(\theta_t)=\frac{1}{n}\sum_{i=1}^n \delta_{z_i},\quad z_i\sim D(\theta_t)$$
> where $D(\theta_t)$ is a **discrete but model-dependent distribution**. The finite-sum form $\frac{1}{n}\sum_{i=1}^n \ell(\theta; z_i)$ is therefore the **exact risk** over the current population at round $t$ and does not break the data-dependence assumption. Moreover, Wasserstein-1 distance and $\epsilon$-sensitivity remain valid for discrete measures, so the assumptions are not broken. For instance, in the **credit-scoring** example of our paper, the population of applicants is finite (individuals who are in the scoring system). Once the model parameter $\theta$ is deployed, individuals strategically modify their features to form a new discrete distribution $D(\theta)$. This setting naturally matches the finite-population assumption.
>
> Finally, we already discuss the **infinite-population** extension in Appendix G. There, we show how SPRINT can be adapted by (a) using mini-batch snapshots (then a bounded-variance or growth condition is needed) or (b) employing a control-variate scheme (Jothimurugesan et al., 2018) that maintains variance control without additional assumptions.
>
>
> > **W2:** “The problem is verbatim the same as those studied in RL, optimal control, and variational inference; please explain why similar convergence results are missing there.”
>
> **Answer:**
>
> We agree with there are some conceptual overlap but also hope to emphasize why performative prediction (PP) becomes a distinct field.
>
> - Relation to reinforcement learning (RL)
>
> Indeed, PP can be reframed as an RL problem as $J(\pi)=E_{\tau\sim p_\pi}[R(\tau)]$. However, we lose the opportunities to do tractable optimization only considering performativity (i.e., model-dependent distribution shifts). While RL optimizes $J(\pi) = E_{\tau \sim p_\pi}[R(\tau)]$ directly under a distribution $p_\pi$ induced by the current policy, it is reward-based and does **not seek an explicit equilibrium** between the policy and the induced distribution. PP defines a *performative stable solution* satisfying $\|\nabla_\theta J(\theta;\theta)\| \le \delta$, which enforces an **equilibrium consistency** between the model parameters and the data distribution they induce.  We also hope to note that in the first work of PP [Perdomo et al., 2021], the authors already addressed the essence of PP compared to RL: *"by studying performative prediction problems within such a broad framework, we lose many of the intricacies of performativity which make the problem interesting and tractable to analyze."*
>
>
> - Relation to optimal control
>
> Optimal control studies dynamic state transitions $x_{t+1}=f(x_t,u_t)$ and minimizes cumulative cost $\sum_t c(x_t,u_t)$. The dependency is primarily temporal, while in PP the dependency is model-dependent and we still aim to find an equilibirum. Optimal control relies on Bellman recursion and dynamic programming; PP instead analyzes convergence to a stationary equilibrium via Wasserstein continuity and Lyapunov functions.
>
> - Modified related work section
>
> **We add a paragraph in related work** to discuss all the above things and review related work, including the ones you mentioned. Please see the modified draft.
>
> > **W3:** “The claim of variance-independent convergence is over-emphasized; Lipschitzness already bounds gradient variance.”
>
> **Answer:**
> We agree that Lipschitzness implicitly limits gradient magnitudes. We say "variance-independent" because the main theoretical result (Thm. 5.4) achieves an $O(1/T)$ rate and an error neighborhood $O(\epsilon^2+\varepsilon^4)$ **without invoking Assumption 3.4** (bounded stochastic variance).  Previous non-convex PP analyses (e.g., Li & Wai 2024) required explicit constants $\sigma_0,\sigma_1$ and derived bounds scaling with them. Our proof eliminates these parameters entirely, and the term *variance-independent* means no explicit dependence on variance constants and no bounded-variance assumption.
>
> To avoid ambiguity, we have rephrased the relevant sentences in the abstract and conclusion as: "*Our rate removes the explicit bounded-variance assumption and contains no variance parameters $\sigma_0,\sigma_1$*" in the modified draft.

---

> > ### Author Response · Authors · 2025-11-19
> >
> > > **Minor points**
> >
> > 1. Thanks for pointing out. **We already modified Assumption 3.1 in the modified draft**.
> >
> > 1. Lemma 5.1 uses Assumption 3.1, 3.3 and $\epsilon$-sensitivity; lemma 5.2 relies on Assumption 3.3 and $\epsilon$-sensitivity. None of the Lemmas use Assumption 3.4 (this serves as a contribution compared to previous work). **We already added all details in the proofs of modified draft.**
> >
> >
> > 2. Regarding the definition of **SPS**, we kindly note that you might have some misunderstanding on the repeated gradient descent procedure of PP setting. Specifically, In standard optimization, we minimize a fixed-distribution objective  $\min_\theta E_{z\sim D}[\ell(\theta;z)]$, so a $\delta$-stationary point satisfies $\|\nabla_\theta E_{z\sim D}[\ell(\theta;z)]\|\le\delta$ under a *static* distribution $D$. In PP each deployment of $\theta_t$ induces a new distribution $D(\theta_t)$ that alters the population (and it can be discrete as shown in the response to W1); hence, the next training round observes data drawn from **a different distribution**. Our objective at $t+1$ is therefore $J(\theta;\theta_t)=E_{z\sim D(\theta_t)}[\ell(\theta;z)]$, and a *performative stable solution* satisfies $\|\nabla_\theta J(\theta;\theta)\|\le\delta$ which means the gradient norm is small when the deployed model is $\theta$ itself, demonstrating the minimal movement during the next round. Although the same symbol $\theta$ appears twice, the two arguments play **distinct roles**:  the first controls the model parameters used for prediction, while the second determines the distribution induced by deploying those parameters. The equilibrium condition $J(\theta;\theta)$ thus represents a **fixed point** of the deployment–distribution feedback loop, which has no analogue in classical ERM or SGD. You can also refer to Li & Wai (2024) for SPS.
> >
> > > Q1: Your experiments report training accuracy. Is there a corresponding notion of test accuracy here for performative prediction? I'm guessing there is no such notion. Which make me think that online learning is a better paradigm for this problem.
> >
> >
> > **Answer:** The concept of “test accuracy” does not directly apply to the performative prediction framework. Our work studies the optimization dynamics and theoretical convergence of models whose deployment continuously reshapes the data distribution $D(\theta)$. In this setting, there is no fixed test distribution, where evaluation naturally concerns the post-deployment accuracy which we already report. Also, we do hope to emphasize our contribution is theoretical rather than empirical benchmarking; extending these ideas to online or streaming environments would constitute a distinct research direction.

---

### Official Review · Reviewer_tZMk · 2025-10-31

**Soundness:** 3
**Presentation:** 3
**Contribution:** 3
**Rating:** 6
**Confidence:** 3

**Summary:**

Strengths
1. Innovative integration of variance reduction into performative prediction: This paper is the first to systematically introduce variance reduction techniques into the performative prediction framework, extending the convergence theory to the non-convex setting.

2. Strong theoretical guarantees under nonconvex settings: Theoretical analysis establishes an O(1/T) convergence rate without assuming bounded gradient variance, representing a major improvement over prior stochastic PP methods which only achieved O(1/\sqrt{T}) rates with variance-dependent error bounds.

3. Empirical results align with theory: Experiments on both tabular (credit scoring) and vision (CIFAR-10, MNIST) datasets show consistent improvements in convergence speed and stability across varying performativity strengths, supporting the theoretical claims.

Weaknesses

1. Experimental scope is narrow: Evaluations are limited to small-scale networks and datasets. The method’s scalability and robustness under more realistic, high-dimensional or large-data performative environments remain untested.

2. Complexity results lack empirical confirmation: The complexity analysis is solid theoretically, but the paper does not include runtime or gradient-call comparisons to validate practical efficiency gains.

**Strengths:**

see Summary

**Weaknesses:**

see Summary

**Questions:**

NA

---

> ### Author Response · Authors · 2025-11-19
> **Response to Reviewer tZMk**
>
> Thanks for your endorsement. We answer your questions as follows:
>
> > W1: Experimental scope is narrow
>
> Regarding the empirical evaluations, we add the following experients in modified draft:
>
> - Error bars using different seeds for partial CIFAR-10 and Credit dataset
>
> we added error bars using different random seeds for Credit and CIFAR-10 (10% data randomly selected) by rerunning the experiments on 3 different seeds. Results are in **Figure 3 and 4, Appendix F** of the modified draft.
>
> - MNIST experiments with different learning rates
>
> We provide an additional experiment in MNIST data with different learning rates and compare the loss/accuracy dynamics of SPRINT and SGD-GD in **Figure 7, Appendix F** of the modified draft. The results demonstrate SPRINT outperfoms SGD-GD under different learning rates.
>
>
> > W2: Complexity results
>
> In Corollary 5.5 and Appendix H, we have derived the IFO complexity of SPRINT to be $\mathcal{O}(\frac{(n^{\alpha} + n^{1+\frac{\alpha}{2}})\Delta_0}{\delta})$. By contrast, the complexity of SGD-GD is $\mathcal{O}(\frac{1}{\delta^2})$, which means there is a trade-off between population size $n$ and the error bound in additional to the error neighborhood $\delta$. When the population size is very large while the error $\delta$ is not too small, the computation of SPRINT can be slower than SGD-GD. **We have added this sentence to Appendix H in modified draft**. As a consequence, the empirical results regarding the gradient call comparisons can just be calculated manually: given that we have $T$ rounds and $m$ iterations each round, the total calls $= T + (\lceil T / m \rceil) \cdot n$.

---

### Official Review · Reviewer_vQs5 · 2025-11-01

**Soundness:** 3
**Presentation:** 3
**Contribution:** 2
**Rating:** 4
**Confidence:** 4

**Summary:**

This work considers the setting of Performative Prediction (PP), a novel risk minimization framework proposed by [Perdomo et al, 2021]. Unlike the tradietional ERM setting, PP assumes that the underlying data distribution also depends on the model parameters. This line of work gained much attention in the recent years, and much effort is devoted to finding a performative stable solution using variants of SGD, under convex/nonconvex loss function with bounded variance.

The authors observe that the nonconvex analysis in [Li and Wai., 2024] suffers from a critical reliance on a growth condition for the stochastic noise and an $O(1/\sqrt{T}) $ rate. To overcome this limitation, they propose a new algorithm and incorporate the variance-reduction technique. The authors further show that their method achieves an $O(1/T)$ convergence rate without assuming the previous noise condition.

**Strengths:**

The paper is clearly written and well-structured. The theoretical results appear sound. The proofs are well organized and generally convincing.

**Weaknesses:**

My concern majorly comes from a contribution perspective. The improvements appear to be restricted to a rather narrow setting of performative prediction. Moreover, the incorporation of variance reduction techniques follows a fairly standard approach. As a result, the paper does not seem to offer many new insights for the broader optimization community. Unless the authors can present more decisive evidence on the significance of their work, the paper in its current form falls short of the expectations for an ICLR publication.

**Questions:**

No further questions.

**Details Of Ethics Concerns:**

This work is purely theoretical and has no negative ethical concerns.

---

> ### Author Response · Authors · 2025-11-19
> **Response to Reviewer vQs5**
>
> Thanks for your endorsement of our theoretical contribution. We address the weakness you mentioned as follows:
>
> > The paper does not seem to offer many new insights for the broader optimization community. Unless the authors can present more decisive evidence on the significance of their work, the paper in its current form falls short of the expectations for an ICLR publication.
>
> - Firstly, we hope to reiterate the significance of performative prediction (PP) settings in real-world applications. In many deployed systems in high-stakes settings, the act of prediction itself changes the data distribution, and the PP framework formally captures this model-distribution feedback loop and thus becomes essential for both correct modelling and robust deployment. Some examples include:
>
> (i) In credit scoring or loan application systems, a model flags certain applicants as high-risk resulting in denial. As applicants observe and respond to the policy (i.e., change their profiles) and the applicant distribution changes;
>
> (ii) In recommendation systems or retention platforms, the model’s output influences user behaviour (what content they consume, how they engage), which in turn alters the next round of user data;
>
> (iii) In public policy or social programmes, predictive models that allocate resources (e.g., subsidy) alter participants' behaviors and lead to different population distributions.
>
> Due to this feedback, optimization and learning methods that assume a static distribution may fail or lead to unintended, unstable outcomes when applied naively. The PP setting offers well-defined equilibrium notions (PS, SPS) and calls for algorithms that explicitly handle the model-distribution interaction. Thus, PP is a foundational framework for studying learning systems where predictions actively influence outcomes and reshape future data. This perspective resonates strongly with ICLR’s growing focus on AI for social good and is a worthwhile presentation.
>
> - Secondly, we hope to emphasize our contribution to the PP framework. **Our core contribution lies in adapting variance reduction (VR) techniques to the performative prediction (PP) problems.** We note that adapting VR techniques in PP is highly non-trivial. Variance reduction techniques are foundational and highly effective in stochastic optimization to avoid the intractable bounded variance assumptions. It has been developed for improving convergence stability and efficiency for optimization in high-variance learning settings. Generalizing VR techniques to PP is particularly important, since PP captures learning scenarios where model deployment affects data distribution, such that stochastic gradients are even noisier due to distributional feedback. This makes VR not only relevant but even more essential for practical and scalable optimization in PP.
>
>
> Regarding the significance of PP and the contribution of our work to PP, we sincerely hope you can reassess the rating.

---

### Author Response · Authors · 2025-11-28
**Rebuttal Summary**

We sincerely thank the extra efforts by the AC to review our paper. In this global rebuttal, we hope to provide a clear outline of how we address the key concerns and update our paper:

> Contribution

Reviewer vQs5 endorsed our theoretical contribution but had some questions on the significance. We summarize the significance of our work as follows:

- **Performative Prediction (PP) is critical in real-world applications**

In many deployed systems in high-stakes settings, the act of prediction itself changes the data distribution and PP framework formally captures this model-distribution feedback loop and thus becomes essential for both correct modelling and robust deployment. Some examples include:

(i) In credit scoring or loan application systems, a model flags certain applicants as high-risk resulting in denial. As applicants observe and respond to the policy (i.e., change their profiles) and the applicant distribution changes;

(ii) In recommendation systems or retention platforms, the model’s output influences user behaviour (what content they consume, how they engage), which in turn alters the next round of user data;

(iii) In public policy or social programmes, predictive models that allocate resources (e.g., subsidy) alter participants' behaviors and lead to different population distributions.

- **Our contribution is crtical to optimization in PP**

Our core contribution lies in adapting variance reduction (VR) techniques to the performative prediction (PP) problems. We note that adapting VR techniques in PP is highly non-trivial. Variance reduction techniques are foundational and highly effective in stochastic optimization to avoid the intractable bounded variance assumptions. It has been developed for improving convergence stability and efficiency for optimization in high-variance learning settings. Generalizing VR techniques to PP is particularly important, since PP captures learning scenarios where model deployment affects data distribution, such that stochastic gradients are even noisier due to distributional feedback. This makes VR not only relevant but even more essential for practical and scalable optimization in PP.

> Finite sum setting is applicable and realistic to PP

Reviewer Dyhx claims that *"finite sum breaks the data dependence assumption"* which is a **factual misunderstanding**. We kindly note that a discrete distribution can still be model-dependent as shown below:

$$D(\theta_t)=\frac{1}{n}\sum_{i=1}^n \delta_{z_i},\quad z_i\sim D(\theta_t)$$
where $D(\theta_t)$ is a **discrete but parameter-dependent distribution**. The finite-sum form $\frac{1}{n}\sum_{i=1}^n \ell(\theta; z_i)$ is therefore the **exact risk** over the current population at round $t$ and does not break the data-dependence assumption. Moreover, Wasserstein-1 distance and $\epsilon$-sensitivity remain valid for discrete measures, so the assumptions are not broken.

For instance, in the **credit-scoring** example of our paper, the population of applicants is finite (individuals who are in the scoring system). Once the model parameter $\theta$ is deployed, individuals strategically modify their features to form a new discrete distribution $D(\theta)$. This setting naturally matches the finite-population assumption.

Finally, we already discuss the **infinite-population** extension in Appendix G. There we show how SPRINT can be adapted by (a) using mini-batch snapshots (then a bounded-variance or growth condition is needed) or (b) employing a control-variate scheme (Jothimurugesan et al., 2018) that maintains variance control without additional assumptions.

> Additional experiments

First, we hope to reiterate our main contribution is theoretical. Also, we believe our experiments are more comprehensive than previous literature on PP (e.g., Perdomo et al., 2021, Li & Wai, 2024) which mostly used synthetic dataset, credit/spam dataset without touching on larger CV dataset such as MNIST/CIFAR. Meanwhile, to the best of our knowledge, all important previous literature on PP and SVRG (e.g., Perdomo et al., 2021, Izzo et al., 2021, Li & Wai, 2024) did not perform experiments on complex NN structures such as Resnet / VGG. Thus, we do believe the pre-existing experiments in our paper are sufficient to support all our theoretical results.

To make our results stronger, we provide plenty of additional experiments in our modified draft:

- We added error bars using different random seeds for Credit and CIFAR-10 (10% data was randomly selected) by rerunning the experiments on 3 different seeds. Results are in **Figure 3 and 4, Appendix F** of the modified draft.

 - We **added an additional MNIST experiment with different learning rates** and plotted the loss/accuracy dynamics of SPRINT and SGD-GD in **Figure 7, Appendix F** of the modified draft.

Hope this can be helpful when you evaluate your paper. Thanks again!

---

> ### Author Response · Authors · 2025-11-28
> **Summary of other edits in modified draft**
>
> We also provide an outline of all other modifications besides additional experiments here:
>
> > Related work and positioning
>
>
> We **added a new paragraph in the related work section** to explicitly discuss the connection and differences between performative prediction and related areas such as reinforcement learning, optimal control, and variational inference, and to incorporate the references suggested by the reviewers.
>
> > Clarification of variance-independence claims
>
>   - To avoid ambiguity, we **rephrased the relevant sentences in the abstract and conclusion** to emphasize we remove the explicit bounded-variance assumption and contains no variance parameters $\sigma_0,\sigma_1$.
>
> > Assumptions and proofs
>   - We corrected the typo in Assumption 3.1 in the main text.
>   - We augmented the proofs of Lemma 5.1–5.3 by explicitly stating which assumptions (Assumption 3.1, 3.3, and $\epsilon$ sensitivity) each lemma uses and clarifying that none of them requires the bounded-variance Assumption 3.4.
>
> > Complexity discussion and when SPRINT may underperform
>
> In **Appendix H**, we **added an explicit sentence** discussing scenarios where SPRINT may be slower or less favorable than SGD-GD, i.e., when the population size $n$ is very large while the target error $\delta$ is not extremely small—based on the IFO complexity.
>
> > Minor presentation fixes
>
> We **fixed the inconsistent equation numbering** (e.g., the duplicated Equation 14) throughout the paper in the modified draft.

---

### Meta-Review · Area_Chair_DdEh · 2025-12-11

**Summary:**

Reviewers indicate the following concerns in their reviews:
- The incorporation of the variance reduction trick is standard, and this does not offer new insights
- Finite sum breaks the data dependent assumption.
- The removal of bounded variance assumption is not essential as there is a Lipschitzness assumption, and the bounded variance assumption follows from this Lipschitzness assumption
- The Lipschitzness assumption is a strict assumption
- The experiments only report training loss, leaving the test loss untouched
- Empirical analysis is not quite sufficient, as there is a lack on the hyperparameter selection, a lack of variance reduction verification in practice
- Complexity results lack empirical confirmation
- More related work should be discussed

**Reviewer Concerns:**

In my opinion, the following reviewer concerns are well addressed:
- More related work should be discussed
- Empirical analysis is not quite sufficient (the authors added some experimental results in the rebuttal phase)
- The experiments only report training loss, leaving the test loss untouched

The following reviewer concerns are not well addressed:
- The incorporation of the variance reduction trick is standard, and this does not offer new insights
- The removal of bounded variance assumption is not essential as there is a Lipschitzness assumption, and the bounded variance assumption follows from this Lipschitzness assumption
- The Lipschitzness assumption is a strict assumption
- Complexity results lack empirical confirmation

**Reviewer Scores:**

Reviewer vQs5 is likely to not change his/her score. The reviewer requests decisive evidence on the significance of the work. The authors' response seem to not provide decisive evidence on the challenges of the incroporation of variance reduction trick.

Reviewer tZMk is likely to not change his/her score. The reviewer requests empiricanl verification on the complexity results. However, the authors did not provide such empirical verification.

Reviewer Dyhx is likely to not change his/her score. The reviewer indicated that the emphasis of variance is overplayed, which is indeed the case. Also, the assumption on Lipschitzness implies the bounded variance assumption.

Reviewer fJGn is likely to not change his/her score. The reviewer indicated that the assumption on Lipschitzness is strict. This is indeed the case as the Lipschitzness assumption already implies the bounded variance assumption.

---

### Decision · Program_Chairs · 2026-01-26

Reject